# A Generalization Result for Convergence in Learning-to-Optimize

**Michael Sucker** [1]   **Peter Ochs** [2]

## Abstract

Learning-to-optimize leverages machine learning to accelerate optimization algorithms. While empirical results show tremendous improvements compared to classical optimization algorithms, theoretical guarantees are mostly lacking, such that the outcome cannot be reliably assured. Especially, convergence is hardly studied in learning-to-optimize, because conventional convergence guarantees in optimization are based on geometric arguments, which cannot be applied easily to learned algorithms. Thus, we develop a probabilistic framework that resembles classical optimization and allows for transferring geometric arguments into learning-to-optimize. Based on our new proof-strategy, our main theorem is a generalization result for parametric classes of potentially non-smooth, non-convex loss functions and establishes the convergence of learned optimization algorithms to critical points with high probability. This effectively generalizes the results of a worst-case analysis into a probabilistic framework, and frees the design of the learned algorithm from using safeguards.

## 1. Introduction

Learning-to-optimize utilizes machine learning techniques to tailor an optimization algorithm to a concrete family of optimization problems with similar structure. While this often leads to enormous gains in performance for problems similar to the ones during training, the learned algorithm might completely fail for others. Thus, to have *trustworthy* algorithms, guarantees are needed. In optimization, the best way to show that the algorithm behaves correctly is by proving its convergence to a critical point. In learning-to-optimize,

however, proving convergence is a hard and long-standing problem. This is due to the fact that the problem instances are *functions*, which cannot be observed globally. Rather, the region explored during training is strongly influenced by the chosen initialization and the maximal number of iterations. This begs a fundamental problem for the theoretical analysis:

> It typically prevents the usage of both limits and the mathematical argument of induction.

As convergence is, by definition, tied to the notion of limits, this subtlety prevents proving that the learned algorithm converges. A way to mitigate this problem rather easily is the usage of safeguards: The update step of the algorithm is restricted to such an extent that it can be analyzed similar to a hand-crafted algorithm, *independently* of the training. Yet, this comes at a price:

> Not only the analysis of the learned algorithm, also its performance is restricted and eventually similar to hand-crafted algorithms.

Intuitively, the degradation in performance can be explained by the fact that this approach attempts to directly apply traditional convergence results, which are not well-suited for learning-to-optimize. Instead, we advocate for taking a new perspective, in which we make more use of our greatest advantage compared to traditional optimization:

> We can actually *observe* the algorithm during training.

Thus, we present a new proof-strategy that allows us to derive convergence in learning-to-optimize by means of *generalization*. The core idea is to show that the properties of the trajectory, which are needed to deduce convergence of the algorithm, actually generalize to unseen problems (test cases). To demonstrate this, we combine a general convergence result from variational analysis with a PAC-Bayesian generalization theorem. This results in our main theorem, which is applicable in a (possibly) non-smooth non-convex optimization setup, and lower bounds the probability to observe a trajectory, generated by the learned algorithm, that converges to a critical point of the loss function. Here, we want to emphasize that, while we derive a convergence result for learning-to-optimize, the idea is not restricted to

[1]Department of Mathematics, University of Tübingen, Tübingen, Germany [2]Department of Mathematics and Computer Science, Saarland University, Saarbrücken, Germany. Correspondence to: Michael Sucker <michael.sucker@math.uni-tuebingen.de>, Peter Ochs <ochs@math.uni-saarland.de>.

*Proceedings of the 42$^{nd}$ International Conference on Machine Learning*, Vancouver, Canada. PMLR 267, 2025. Copyright 2025 by the author(s).

optimization. Rather, it applies more generally to sequential prediction models that exhibit a Markovian structure.

## 2. Related Work

This work draws on the fields of learning-to-optimize, the PAC-Bayesian learning approach, and convergence results based on the Kurdyka-Łojasiewicz property. For an introduction to learning-to-optimize, Chen et al. (2022) provide a good overview about the variety of approaches. Similarly, for the PAC-Bayesian approach, good introductory references are given by Guedj (2019), Alquier (2024), and Hellström et al. (2025), and for the usage of the Kurdyka-Łojasiewicz property, we refer to Attouch et al. (2013).

**Learning-to-Optimize with Guarantees.** To date, learned optimization methods show impressive performance, yet lack theoretical guarantees (Chen et al., 2022). However, in some applications convergence guarantees are indispensable: It was shown that learning-based methods might fail to reconstruct the crucial details in a medical image (Möller et al., 2019). In the same work, the authors prove convergence of their learned method by restricting the update to descent directions. Similar safeguarding techniques were employed by Prémont-Schwarz et al. (2022) and Heaton et al. (2023). The basic idea is to constrain the learned object in such a way that known convergence results are applicable, and it has been applied successfully for different schemes and under different assumptions (Sreehari et al., 2016; Chan et al., 2017; Teodoro et al., 2017; Tirer & Giryes, 2019; Buzzard et al., 2018; Ryu et al., 2019; Sun et al., 2019; Terris et al., 2021; Cohen et al., 2021). A major advantage of these "constrained" methods is the fact that the number of iterations is not restricted a priori and that, often, some convergence guarantees can be provided. A major drawback, however, is their severe restriction: Typically, the update-step has to satisfy certain geometric properties, and the results only apply to specific algorithms and/or problems. Another approach, pioneered by Gregor & LeCun (2010), is unrolling, which limits the number of iterations, yet can be applied to any iterative algorithm. Here, the IHT algorithm is studied by Xin et al. (2016) while Chen et al. (2018) consider the unrolled ISTA. However, in the theoretical analysis of unrolled algorithms, the notion of convergence itself is difficult, and one rather has to consider the generalization performance: This has been done by means of Rademacher complexity (Chen et al., 2020), by using a stability analysis (Kobler et al., 2022), or in terms of PAC-Bayesian generalization guarantees (Sucker & Ochs, 2023; Sucker et al., 2024). Recently, generalization guarantees based on the whole trajectory of the algorithm, for example, the expected time to reach the stopping criterion, have been proposed (Sucker & Ochs, 2024). The main drawback of generalization guarantees is their reliance on a specific distribution. To solve this, another line of work studies the design of learned optimization algorithms and their training, and how it affects the possible guarantees (Wichrowska et al., 2017; Metz et al., 2019; 2022). Here, Liu et al. (2023) identify common properties of basic optimization algorithms and propose a math-inspired architecture. Similarly, Castera & Ochs (2024) analyze widely used optimization algorithms, extract common geometric properties from them, and provide design-principles for learning-to-optimize.

**PAC-Bayesian Generalization Bounds.** The PAC-Bayesian framework allows for giving high probability bounds on the risk. The key ingredient is a change-of-measure inequality, which determines the divergence or distance in the resulting bound. While most bounds involve the Kullback–Leibler divergence as measure of proximity (McAllester, 2003a;b; Seeger, 2002; Langford & Shawe-Taylor, 2002; Catoni, 2004; 2007; Germain et al., 2009), more recently other divergences have been used (Honorio & Jaakkola, 2014; London, 2017; Bégin et al., 2016; Alquier & Guedj, 2018; Ohnishi & Honorio, 2021; Amit et al., 2022; Haddouche & Guedj, 2023). In doing so, the PAC-bound relates the true risk to other terms such as the empirical risk. Yet, it does not directly say anything about the absolute numbers. Therefore, one typically aims to minimize the provided upper bound (Langford & Caruana, 2001; Dziugaite & Roy, 2017; Pérez-Ortiz et al., 2021; Thiemann et al., 2017). Nevertheless, a known difficulty in PAC-Bayesian learning is the choice of the prior distribution, which strongly influences the performance of the learned models and the theoretical guarantees (Catoni, 2004; Dziugaite et al., 2021; Pérez-Ortiz et al., 2021). In part, this is due to the fact that the divergence term can dominate the bound, such that the posterior is close to the prior. Especially, this applies to the Kullback-Leibler divergence, and lead to the idea of choosing a data- or distribution-dependent prior (Seeger, 2002; Parrado-Hernández et al., 2012; Lever et al., 2013; Dziugaite & Roy, 2018; Pérez-Ortiz et al., 2021).

**The Kurdyka-Łojasiewicz inequality.** Single-point convergence of the trajectory of an algorithm is a challenging problem, especially in non-smooth non-convex optimization. For example, Absil et al. (2005) show that this might fail even for simple algorithms like gradient descent on highly smooth functions. Further, they show that a remedy is provided by the *Łojasiewicz inequality*, which holds for real analytic functions (Bierstone & Milman, 1988). The large class of tame functions or definable functions excludes many pathological failure cases, and extensions of the Łojasiewicz inequality to smooth definable functions are provided by Kurdyka (1998). Similarly, extensions to the nonsmooth subanalytic or definable setting are shown by Bolte et al. (2007b), Bolte et al. (2007a), and Attouch & Bolte (2009),

which yields the *Kurdyka–Łojasiewicz inequality*. It is important to note that most functions in practice are definable and thus satisfy the Kurdyka–Łojasiewicz inequality automatically. Using this, several algorithms have been shown to converge even for nonconvex functions (Attouch & Bolte, 2009; Attouch et al., 2010; 2013; Bolte et al., 2014; Ochs et al., 2014; Ochs, 2019).

## 3. Contributions

- We present a novel approach for deducing the convergence of a generic learned algorithm with high probability. In doing so, we effectively generalize the results of a worst-case convergence analysis into a probabilistic setting. Furthermore, the methodology does not restrict the design of the algorithm and is widely applicable, that is, it can also be used for other sequential prediction models that exhibit a Markovian structure.

- To showcase the idea, we combine the PAC-Bayesian generalization theorem provided by Sucker & Ochs (2024) with the abstract convergence theorem provided by Attouch et al. (2013) to derive a new convergence result for our learned optimization algorithm on (possibly) non-smooth non-convex loss-functions. In doing so, we bring together highly advanced tools from non-smooth non-convex optimization, stochastic process theory, and PAC-Bayesian learning theory, and effectively solve a long-standing problem of learning-to-optimize, namely how to obtain convergence guarantees *without* limiting the design of the algorithm.

- We conduct two experiments to show the validity of our claims: We use a neural-network based iterative optimization algorithm to a) solve quadratic problems and b) to train another neural network. In both cases, the learned algorithm outperforms the baseline and converges to a critical point with high probability.

## 4. Simplified Key Idea

Before detailing the setup for learning-to-optimize, we shortly (and informally) present the main underlying idea of our proof-strategy, which otherwise might be obscured by the details: Given an object $x$ and properties $a$, $b$, and $c$, we are interested in the implication

$$x \text{ satisfies } a \wedge b \implies x \text{ satisfies } c \,.$$

Whenever a single object $x$ has properties $a$ and $b$, we are *sure* that $x$ also possesses $c$. However, given a collection of objects $\{x_1, x_2, ...\}$, if the properties $a$ and $b$ only hold for some of these objects, the traditional "implication" is invalid for the collection, and the language of probability theory seems more appropriate: Here, $a$, $b$ and $c$ have to be rephrased as *sets* $\mathsf{A} := \{x : x \text{ has property } a\}$, $\mathsf{B} := \{x :$

$x$ has property $b\}$, and $\mathsf{C} := \{x : x \text{ has property } c\}$, such that the implication translates into an inclusion:

$$\mathsf{A} \cap \mathsf{B} = \{x : x \text{ satisfies } a \wedge b\} \subset \{x : x \text{ satisfies } c\} = \mathsf{C} \,.$$

This enables a more fine-grained result: If we are given a probability measure $\mu$ over objects $x$, we can always conclude that $\mu\{\mathsf{A} \cap \mathsf{B}\} \leq \mu\{\mathsf{C}\}$, that is, it is more likely to observe an object $x$ with property $c$ than to observe an object with properties $a$ and $b$. Furthermore, if $\mu\{\mathsf{A} \cap \mathsf{B}\} = 1$, we deduce that $c$ holds *almost surely*.

Most of the time, however, calculating $\mu\{\mathsf{A} \cap \mathsf{B}\}$ is infeasible, so that it needs to be estimated on a data set. In this case, two questions arise:

(i) Is the estimate *representative* for unseen data?

(ii) Why do we not simply estimate $\mu\{\mathsf{C}\}$ directly?

The first question can be answered in terms of a generalization result. By contrast, the second question can be more subtle: *If* the property $c$ is observable, estimating $\mu\{\mathsf{C}\}$ should be preferred. However, this is not always possible. In our case, for example, the objects $x$ will be whole sequences, the property $c$ will be convergence to a critical point, and $\mu$ will be the distribution of a Markov process generated by the algorithm. Thus, without further assumptions it is practically impossible to observe property $c$ directly, because convergence of a sequence is a so-called *asymptotic event*, which belongs to the tail-$\sigma$-algebra, that is, it does not depend on any finite number of iterates and therefore cannot be observed.

To summarize: Ultimately, we are interested in how likely it is to observe an object $x$ that possesses property $c$ (here: a sequence generated by the learned algorithm that converges to a stationary point). For this, we need at least the distribution $\mu$ of the objects $x$ under consideration (see Theorem 6.3). Additionally, since property $c$ is unobservable, we resort to properties $a$ and $b$, which imply $c$ (see Theorem 6.5). Then, to be able to assign probabilities to these properties, we need to translate them into measurable sets in the appropriate space (see Section 7.1). This allows for estimating the probability to observe objects $x$ that possess $a$ and $b$, which in turn is a lower bound on how many objects possess $c$. Finally, since this estimate depends on the training data, we need to make sure that it also generalizes to unseen problems (see Theorem 7.6).

## 5. Notation

We write generic sets in type-writer font, for example, $\mathsf{A} \subset \mathbb{R}^d$, and generic spaces in script-font, for example, $\mathscr{X}$. Given a metric space $\mathscr{X}$, $\mathsf{B}_\varepsilon(x)$ denotes the open ball

around $x \in \mathcal{X}$ with radius $\varepsilon > 0$, and we assume every metric space to be endowed with the metric topology and corresponding Borel $\sigma$-field $\mathfrak{B}(\mathcal{X})$. Similarly, given a product space $\mathcal{X} \times \mathcal{Y}$, the product $\sigma$-algebra is denoted by $\mathfrak{B}(\mathcal{X}) \otimes \mathfrak{B}(\mathcal{Y})$. We consider the space $\mathbb{R}^d$ with Euclidean norm $\|\cdot\|$ and, for notational simplicity, abbreviate $\mathcal{X} := \mathbb{R}^d$ and $\mathscr{P} := \mathbb{R}^q$. The space of sequences in $\mathcal{X}$ is denoted by $\mathcal{X}^{\mathbb{N}_0}$, and we endow it with the product $\sigma$-algebra, which is the smallest $\sigma$-algebra, such that all canonical projections $\pi_i : \mathcal{X}^{\mathbb{N}_0} \to \mathcal{X}$, $(z^{(t)})_{t \in \mathbb{N}_0} \mapsto z^{(i)}$, are measurable. For notions from non-smooth analysis, we follow Rockafellar & Wets (1998). In short, a function $f : \mathcal{X} \to \mathbb{R} \cup \{+\infty\}$ is called *proper*, if $f(z) < +\infty$ for at least one $z \in \mathcal{X}$, and we denote its *effective domain* by $\operatorname{dom} f$. Further, it is called *lower semi-continuous*, if $\liminf_{z \to \bar{z}} f(z) \geq f(\bar{z})$ for all $\bar{z} \in \mathcal{X}$. Furthermore, for $z \in \operatorname{dom} f$, the (limiting) *subdifferential* of $f$ at $z$ is denoted by $\partial f(z)$. Similarly, for $f : \mathcal{X} \times \mathscr{P} \to \mathbb{R} \cup \{+\infty\}$, $\partial_1 f(z_1, z_2)$ denotes the (partial) subdifferential of $f(\cdot, z_2)$ at $z_1$. In general, $\partial f : \mathcal{X} \rightrightarrows \mathcal{X}$ is a set-valued mapping, and we denote its *domain* and *graph* by $\operatorname{dom} \partial f$ and $\operatorname{gph} \partial f$, respectively. Here, a set-valued mapping $T : \mathbb{R}^k \rightrightarrows \mathbb{R}^l$ is said to be *outer semi-continuous* at $\bar{x}$, if $\limsup_{x \to \bar{x}} T(x) \subset T(\bar{x})$, where the *outer limit* is defined as $\limsup_{x \to \bar{x}} T(x) = \{u \mid \exists x^{(t)} \to \bar{x}, \exists u^{(t)} \to u \text{ with } u^{(t)} \in T(x^{(t)})\}$. For convenience of the reader, we have collected more details in Appendix A. Also, the exact definition of a *Kurdyka-Łojasiewicz* (KL) function can be found there, as it is quite intricate. However, for the following, it is actually sufficient to understand that these are functions that are "sharp up to reparametrization" (Attouch et al., 2013), and that many functions encountered in real-world problems are KL-functions. Finally, the space of measures on $\mathcal{X}$ is denoted by $\mathcal{M}(\mathcal{X})$, and all probability measures that are absolutely continuous w.r.t. a reference measure $\mu \in \mathcal{M}(\mathcal{X})$ are denote by $\mathcal{M}_1(\mu) := \{\nu \in \mathcal{M}(\mathcal{X}) : \nu \ll \mu \text{ and } \nu[\mathcal{X}] = 1\}$. Here, the Kullback-Leiber divergence between two measures $\mu$ and $\nu$ is defined as $D_{\mathrm{KL}}(\nu \parallel \mu) = \nu[\log(f)] = \int_{\mathcal{X}} \log(f(x)) \, \nu(dx)$, if $\nu \ll \mu$ with density $f$, and $+\infty$ otherwise.

## 6. Problem Setup

Instead of considering a whole class of problems, we assume that we are given a *parametric loss-function* $\ell : \mathcal{X} \times \mathscr{P} \to [0, \infty]$ and a random variable $P$ taking values in the parameter space $\mathscr{P} = \mathbb{R}^q$. Here, $\mathcal{X} = \mathbb{R}^d$ is the optimization space and the ultimate goal would be to solve

$$\min_{z \in \mathcal{X}} \ell(z, p)$$

for every realization $P = p$. Since we include non-convex optimization problems, finding the global minimum is infeasible, and we focus on finding a critical point instead. For this, we apply an *algorithmic update* $\mathcal{A} : \mathcal{H} \times \mathscr{P} \times \mathcal{X} \times \mathcal{R} \to$

$\mathcal{X}$ iteratively to the initial state $z^{(0)} \in \mathcal{X}$:

$$z^{(t+1)} = \mathcal{A}(h, p, z^{(t)}, r^{(t+1)}), \quad t \in \mathbb{N}_0. \quad (1)$$

Here, the *hyperparameters* $h \in \mathcal{H}$ allow for adjusting the algorithm, the *parameters* $p \in \mathscr{P}$ specify the loss function $\ell(\cdot, p)$ the algorithm is applied to, and $r^{(t+1)} \in \mathcal{R}$ models the *(internal) randomness* of the algorithm. To find suitable hyperparameters $h \in \mathcal{H}$, the algorithm $\mathcal{A}$ is trained on an i.i.d. data set of problem parameters $S = (P_1, ..., P_N)$ in such a way that its performance on $\ell$ is superior to that of traditional algorithms. However, in optimization "performance" is usually measured based on the whole sequence $(z^{(t)})_{t \in \mathbb{N}_0}$, for example, a linear rate of convergence has to hold for all iterations. Thus, to deal with such measures of performance, one needs to access the trajectories generated by $\mathcal{A}$. This is where the Markovian model of Sucker & Ochs (2024) comes into play: If $h$ and $p$ are fixed along the iterations, Equation (1) can be read as the *functional equation* of a Markov process $\xi = (z^{(t)})_{t \in \mathbb{N}_0}$, which defines a distribution on $\mathcal{X}^{\mathbb{N}_0}$ and thus allows for analyzing these trajectories. It is based on the following two assumptions:

**Assumption 6.1.** The state space $(\mathcal{X}, \mathfrak{B}(\mathcal{X}), \mathbb{P}_I)$, the parameter space $(\mathscr{P}, \mathfrak{B}(\mathscr{P}), \mathbb{P}_P)$, the hyperparameter space $(\mathcal{H}, \mathfrak{B}(\mathcal{H}), \mathbb{P}_H)$, and the randomization space $(\mathcal{R}, \mathfrak{B}(\mathcal{R}), \mathbb{P}_R)$ are Polish[1] probability spaces.

**Assumption 6.2.** The (possibly extended-valued) *loss-function* $\ell : \mathcal{X} \times \mathscr{P} \to [0, \infty]$ and the *algorithmic update* $\mathcal{A} : \mathcal{H} \times \mathscr{P} \times \mathcal{X} \times \mathcal{R} \to \mathcal{X}$ are both measurable.

Then, Sucker & Ochs (2024) construct a suitable probability space $(\Omega, \mathfrak{A}, \mathbb{P})$ that correctly models the joint distribution of $(H, P, \xi)$ on $\mathcal{H} \times \mathscr{P} \times \mathcal{X}^{\mathbb{N}_0}$, that is, the joint distribution over hyperparameters $h$, problem parameters $p$, and corresponding trajectories $\xi$ generated by $\mathcal{A}(h, p, \cdot, \cdot)$. By leveraging a well-known result due to Catoni (2007), they show that properties of trajectories $\xi$, *encoded as sets* $\mathsf{A} \in \mathfrak{B}(\mathscr{P}) \otimes \mathfrak{B}(\mathcal{X})^{\otimes \mathbb{N}_0}$, generalize in a PAC-Bayesian way (Sucker & Ochs, 2024, Theorem 42):

**Theorem 6.3.** *Let* $\mathsf{A} \subset \mathscr{P} \times \mathcal{X}^{\mathbb{N}_0}$ *be measurable, and define* $\Phi_a^{-1}(p) := \frac{1 - \exp(-ap)}{1 - \exp(-a)}$. *Then, for* $\lambda \in (0, \infty)$*, it holds that:*

$$\mathbb{P}_S \Big\{ \forall \rho \in \mathcal{M}_1(\mathbb{P}_H) : \rho[\mathbb{P}_{(P, \xi) \mid H} \{\mathsf{A}\}] \leq$$

$$\Phi_{\frac{\lambda}{N}}^{-1} \Big( \frac{1}{N} \sum_{n=1}^N \rho \left[ \mathbb{P}_{(P_n, \xi_n) \mid H, P_n} \{\mathsf{A}\} \right]$$

$$+ \frac{D_{\mathrm{KL}}(\rho \parallel \mathbb{P}_H) + \log\left(\frac{1}{\varepsilon}\right)}{\lambda} \Big) \Big\} \geq 1 - \varepsilon.$$

Here, $\mathbb{P}_H$ is the so-called *prior* over hyperparameters. Every $\rho \in \mathcal{M}_1(\mathbb{P}_H)$ is called a *posterior*, $\mathbb{P}_{(P, \xi) \mid H}$ is the conditional distribution of the parameters with corresponding

---

[1]A *Polish space* is a separable topological space that admits a complete metrization.

trajectory, given the hyperparameters, and $\mathbb{P}_S$ is the distribution of the data set $S$. On an intuitive level, the probability to observe a problem instance $\ell(\cdot, p)$ and a corresponding trajectory $\xi$ generated by the algorithm $\mathcal{A}(h, p, \cdot, \cdot)$, which satisfies the properties encoded in A, can be bounded based on empirical estimates. It is important to note that, except for a brief remark, Sucker & Ochs (2024) do not make any further use of this result. In this paper, we observe the power of this theorem and apply it to the set of sequences that converge to a critical point of $\ell$.

**Definition 6.4.** Let $f : \mathcal{X} \to \mathbb{R} \cup \{+\infty\}$ be proper. A point $z \in \mathcal{X}$ is called *critical* for $f$, if $0 \in \partial f(z)$.

It is crucial to realize that, usually it is *impossible* to observe convergence directly, as it belongs to the class of *tail events*, which do not depend on any finite number of iterates. Hence, to be able to apply the generalization result from above, we need abstract properties that do not depend on the implementation of the algorithm, are easily observable during training, and are sufficient to deduce convergence, which, as we discussed in the related work, is highly non-trivial in the challenging setup of non-smooth non-convex optimization. Nevertheless, such conditions are provided by the following result due to Attouch et al. (2013, Theorem 2.9):

**Theorem 6.5.** *Let $f : \mathcal{X} \to \mathbb{R} \cup \{+\infty\}$ be a proper lower semi-continuous function that is bounded from below. Further, suppose that $(z^{(t)})_{t \in \mathbb{N}_0} \subset \mathcal{X}$ is a sequence satisfying the following property: There exist positive scalars $a$ and $b$, such that the following conditions hold:*

(i) *Sufficient-decrease: For each $t \in \mathbb{N}_0$, $f(z^{(t+1)}) + a\|z^{(t+1)} - z^{(t)}\|^2 \leq f(z^{(t)})$.*

(ii) *Relative-error: For each $t \in \mathbb{N}_0$, there exists $v^{(t+1)} \in \partial f(z^{(t+1)})$ with $\|v^{(t+1)}\| \leq b\|z^{(t+1)} - z^{(t)}\|$.*

(iii) *Continuity: For any convergent subsequence $z^{(t_j)} \overset{j \to \infty}{\to} \hat{z}$, we have $f(z^{(t_j)}) \overset{j \to \infty}{\to} f(\hat{z})$.*

*If, additionally, the sequence $(z^{(t)})_{t \in \mathbb{N}_0}$ is bounded and $f$ is a Kurdyka-Łojasiewicz function, then $(z^{(t)})_{t \in \mathbb{N}_0}$ converges to a critical point of $f$.*

*Remark* 6.6. (i) The continuity condition cannot be checked in practice. Therefore, we will have to assume that $\ell(\cdot, p)$ is continuous on its domain.

(ii) Theorem 2.9 of Attouch et al. (2013) is stated slightly different: They assume existence of a convergent subsequence instead of boundedness. Yet, boundedness implies existence and is standard (Bolte et al., 2014).

(iii) In Appendix B, we provide examples to underline the necessity of these conditions. Especially, we show that the sufficient-descent condition alone is not sufficient

for deducing convergence to a critical point, which appears to be a common misconception.

Since we want to employ these results from variational calculus, we have to make the restriction to $\mathcal{X} = \mathbb{R}^d$ and $\mathcal{P} = \mathbb{R}^q$. However, one could also consider a (finite-dimensional) state space $\mathcal{X}$ that *encompasses* the space of the optimization variable, that is, $\mathcal{X} = \mathbb{R}^{d_1} \times \mathbb{R}^{d_2}$, and, by projecting onto $\mathbb{R}^{d_1}$, the results carry over immediately.[2]

**Assumption 6.7.** We have $\mathcal{X} = \mathbb{R}^d$ and $\mathcal{P} = \mathbb{R}^q$, and the function $\ell : \mathcal{X} \times \mathcal{P} \to [0, \infty]$ is proper, lower semi-continuous, and continuous on $\mathrm{dom}\, \ell$. Furthermore, the map $(z, p) \mapsto \partial_1 \ell(z, p)$ is outer semi-continuous.

# 7. Theoretical Results

Now, we concretize our key idea from Section 4 for the setting of learning-to-optimize, and combine Theorem 6.3 with Theorem 6.5 to get a generalization result for the convergence of learned algorithms to critical points. In doing so, we bring together advanced tools from learning theory and optimization: We show that the probability to observe a parameter $p$ and a corresponding trajectory $\xi$, which converges to a critical point of $\ell(\cdot, p)$, generalizes. For this, we formulate the sufficient-descent condition, the relative-error condition, and the boundedness assumption as *measurable* sets in $\mathcal{P} \times \mathcal{X}^{\mathbb{N}_0}$, such that their intersection is exactly the set of sequences satisfying the properties of Theorem 6.5.

## 7.1. Measurability

While measurability is usually dismissed as a technicality, it is absolutely necessary for the validity of employed theorems. Thus, to be able to apply Theorem 6.3, we have to show that these sets are actually measurable w.r.t. $\mathfrak{B}(\mathcal{P}) \otimes \mathfrak{B}(\mathcal{X})^{\otimes \mathbb{N}_0}$, which, unfortunately, is not a given. Hence, denote the (parametric) set of critical points of $\ell$ by

$$\mathsf{A}_{\mathrm{crit}} := \{(p, z) \in \mathcal{P} \times \mathcal{X} \; : \; 0 \in \partial_1 \ell(z, p)\}.$$

Then, the section $\mathsf{A}_{\mathrm{crit},p} := \{z \in \mathcal{X} \; : \; (p, z) \in \mathsf{A}_{\mathrm{crit}}\}$ is the set of critical points of $\ell(\cdot, p)$.

**Lemma 7.1.** *Suppose that Assumptions 6.1, 6.2 and 6.7 hold. Define the (parametric) set of sequences that converge to a critical point of $\ell$ as*

$$\mathsf{A}_{\mathrm{conv}} := \{(p, (z^{(t)})_{t \in \mathbb{N}_0}) \in \mathcal{P} \times \mathcal{X}^{\mathbb{N}_0} \; :$$
$$\exists z^* \in \mathsf{A}_{\mathrm{crit},p} \; s.t. \; \lim_{t \to \infty} \|z^{(t)} - z^*\| = 0\}.$$

*Then $\mathsf{A}_{\mathrm{conv}}$ is measurable.*

*Proof.* The proof is highly non-trivial and can be found in Appendix C. However, the idea is simple: $\mathsf{A}_{\mathrm{conv}}$ can be

---

[2]For the cost of an even heavier notation, which is why we have omitted doing it here.

written as *countable* intersection/union of measurable sets. This is possible, because we consider Polish spaces, that is, they have a countable dense subset and they are complete, that is, limits of Cauchy sequences are inside the space. □

**Lemma 7.2.** *Assume that Assumptions 6.1 and 6.2 hold. Define the (parametric) set of sequences that satisfy the* sufficient-descent condition *as*

$$\mathsf{A}_{\mathrm{desc}} := \Big\{ (p, (z^{(t)})_{t\in\mathbb{N}_0}) \in \mathscr{P} \times \mathfrak{X}^{\mathbb{N}_0} :$$
$$(z^{(t)})_{t\in\mathbb{N}_0} \subset \mathrm{dom}\,\ell(\cdot, p) \text{ and } \exists a > 0 \text{ s.t. } \forall t \in \mathbb{N}_0$$
$$\ell(z^{(t+1)}, p) + a\|z^{(t+1)} - z^{(t)}\|^2 \leq \ell(z^{(t)}, p) \Big\}.$$

*Then $\mathsf{A}_{\mathrm{desc}}$ is measurable.*

*Proof.* The proof can be found in Appendix D. □

We proceed with the relative error condition. It involves a union over all subgradients, and thus might not be measurable. Hence, we have to restrict to subgradients given through a *measurable selection*, that is, a measurable function $v : \mathrm{dom}\,\partial_1\ell \to \mathfrak{X}$, such that $v(z, p) \in \partial_1\ell(z, p)$ for every $(z, p) \in \mathrm{dom}\,\partial_1\ell$. Under the given assumptions, its existence is guaranteed by Corollary E.3.

**Lemma 7.3.** *Suppose that Assumptions 6.1, 6.2 and 6.7 hold. Define the (parametric) set of sequences that satisfy the* relative-error condition *as*

$$\mathsf{A}_{\mathrm{err}} := \Big\{ (p, (z^{(t)})_{t\in\mathbb{N}_0}) \in \mathscr{P} \times \mathfrak{X}^{\mathbb{N}_0} :$$
$$(p, z^{(t)}) \in \mathrm{dom}\,\partial_1\ell \; \forall t \in \mathbb{N}_0 \text{ and } \exists b > 0 \text{ s.t. } \forall t \in \mathbb{N}_0$$
$$\|v(z^{(t+1)}, p)\| \leq b\|z^{(t+1)} - z^{(t)}\| \Big\}.$$

*Then $\mathsf{A}_{\mathrm{err}}$ is measurable.*

*Proof.* The proof can be found in Appendix E. □

**Lemma 7.4.** *Assume that Assumption 6.1 holds. Define the set of* bounded sequences *as:*

$$\tilde{\mathsf{A}}_{\mathrm{bound}} = \Big\{ (z^{(t)})_{t\in\mathbb{N}_0} \in \mathfrak{X}^{\mathbb{N}_0} :$$
$$\exists c \geq 0 \text{ s.t. } \|z^{(t)}\| \leq c \; \forall t \in \mathbb{N}_0 \Big\}.$$

*Then $\mathsf{A}_{\mathrm{bound}} := \mathscr{P} \times \tilde{\mathsf{A}}_{\mathrm{bound}}$ is measurable.*

*Proof.* The proof can be found in Appendix F. □

### 7.2. Convergence to critical points

We are now in a position to derive our main result.

**Corollary 7.5.** *Suppose that Assumptions 6.1, 6.2, and 6.7 hold. Furthermore, assume that $\ell(\cdot, p)$ is a Kurdyka-Łojasiewicz function for every $p \in \mathscr{P}$. Then the sets $\mathsf{A}_{\mathrm{desc}} \cap \mathsf{A}_{\mathrm{err}} \cap \mathsf{A}_{\mathrm{bound}} \subset \mathscr{P} \times \mathfrak{X}^{\mathbb{N}_0}$ and $\mathsf{A}_{\mathrm{conv}} \subset \mathscr{P} \times \mathfrak{X}^{\mathbb{N}_0}$ are measurable, and it holds that:*

$$\mathsf{A}_{\mathrm{desc}} \cap \mathsf{A}_{\mathrm{err}} \cap \mathsf{A}_{\mathrm{bound}} \subset \mathsf{A}_{\mathrm{conv}}.$$

*Proof.* Let $(p, (z^{(t)})_{t\in\mathbb{N}_0}) \in \mathsf{A}_{\mathrm{desc}} \cap \mathsf{A}_{\mathrm{err}} \cap \mathsf{A}_{\mathrm{bound}}$. Thus, $(z^{(t)})_{t\in\mathbb{N}_0}$ satisfies both the sufficient-descent and the relative-error condition for $\ell(\cdot, p)$, and $(z^{(t)})_{t\in\mathbb{N}_0}$ stays bounded. Further, $(z^{(t)})_{t\in\mathbb{N}_0}$ also satisfies the continuity condition, since we have $(z^{(t)})_{t\in\mathbb{N}_0} \subset \mathrm{dom}\,\ell(\cdot, p)$ and $\ell$ is continuous on its domain. Hence, Theorem 6.5 implies that $(z^{(t)})_{t\in\mathbb{N}_0}$ converges to a critical point of $\ell(\cdot, p)$, that is, there exists $z^* \in \mathsf{A}_{\mathrm{crit},p}$, such that $\lim_{t\to\infty} \|z^{(t)} - z^*\| = 0$. Therefore, $(p, (z^{(t)})_{t\in\mathbb{N}_0}) \in \mathsf{A}_{\mathrm{conv}}$. □

In particular, if $\mu$ is a (probability) measure on $\mathscr{P} \times \mathfrak{X}^{\mathbb{N}_0}$, for example, $\mu = \mathbb{P}_{(P,\xi)|H=h}$ for a given $h \in \mathscr{H}$, by the monotonicity of measures it holds that:

$$\mu\{\mathsf{A}_{\mathrm{desc}} \cap \mathsf{A}_{\mathrm{err}} \cap \mathsf{A}_{\mathrm{bound}}\} \leq \mu\{\mathsf{A}_{\mathrm{conv}}\}.$$

This idea yields our main theorem:

**Theorem 7.6.** *Suppose that Assumptions 6.1, 6.2, and 6.7 hold. Further, assume that $\ell(\cdot, p)$ is a Kurdyka–Łojasiewicz function for every $p \in \mathscr{P}$. Abbreviate $\mathsf{A} := \mathsf{A}_{\mathrm{desc}} \cap \mathsf{A}_{\mathrm{err}} \cap \mathsf{A}_{\mathrm{bound}}$. Then, for $\lambda \in (0, \infty)$, it holds that:*

$$\mathbb{P}_S\Big\{ \forall \rho \in \mathcal{M}_1(\mathbb{P}_H) \; : \; \rho[\mathbb{P}_{(P,\xi)|H}\{\mathsf{A}_{\mathrm{conv}}\}] \geq 1-$$
$$\Phi_{\frac{\lambda}{N}}^{-1}\Big( \frac{1}{N}\sum_{n=1}^{N} \rho\left[\mathbb{P}_{(P_n,\xi_n)|H,P_n}\{\mathsf{A}^c\}\right]$$
$$+ \frac{D_{\mathrm{KL}}(\rho \,\|\, \mathbb{P}_H) + \log\left(\frac{1}{\varepsilon}\right)}{\lambda} \Big) \Big\} \geq 1 - \varepsilon.$$

*Proof.* By taking the complementary events in Corollary 7.5, we have $\mathbb{P}_H$-a.s.:

$$\mathbb{P}_{(P,\xi)|H}\{\mathsf{A}^c\} \geq 1 - \mathbb{P}_{(P,\xi)|H}\{\mathsf{A}_{\mathrm{conv}}\} .$$

By Theorem 6.3, for any measurable set $\mathsf{B} \subset \mathscr{P} \times \mathfrak{X}^{\mathbb{N}_0}$ and $\lambda \in (0, \infty)$, we have:

$$\mathbb{P}_S\Big\{ \forall \rho \in \mathcal{M}_1(\mathbb{P}_H) \; : \; \rho[\mathbb{P}_{(P,\xi)|H}\{\mathsf{B}\}] \leq$$
$$\Phi_{\frac{\lambda}{N}}^{-1}\Big( \frac{1}{N}\sum_{n=1}^{N} \rho\left[\mathbb{P}_{(P_n,\xi_n)|H,P_n}\{\mathsf{B}\}\right]$$
$$+ \frac{D_{\mathrm{KL}}(\rho \,\|\, \mathbb{P}_H) + \log\left(\frac{1}{\varepsilon}\right)}{\lambda} \Big) \Big\} \geq 1 - \varepsilon.$$

Hence, using $\mathsf{B} := \mathsf{A}^c$, inserting the inequality above, and rearranging the terms yields the result. □

*Remark* 7.7. (i) The lower bound actually applies to $\mathbb{P}_{(P,\xi)|H}\{\mathsf{A}\}$. Since the difference $\mathbb{P}_{(P,\xi)|H}\{\mathsf{A}_{\mathrm{conv}} \setminus \mathsf{A}\}$ is unknown, we do not know the tightness of this bound for $\mathbb{P}_{(P,\xi)|H}\{\mathsf{A}_{\mathrm{conv}}\}$.

(ii) We want to stress the following: We do not assume that the conditions of Theorem 6.5, for example, the sufficient-descent condition, do hold *per se*. As described in Section 4 about the underlying idea, this theorem is rather about the fact that, based on our observations during training, we can deduce *how often* these conditions will hold on unseen problem instances.

(iii) For large $N$, if $\lambda$ is chosen correctly, we can approximate $\Phi_{\frac{\lambda}{N}}^{-1}(p) \approx p$. Assuming this holds, and abbreviating the empirical approximation as $\hat{\mathbb{P}}_{(P,\xi)|H} := \frac{1}{N}\sum_{n=1}^{N}\mathbb{P}_{(P_n,\xi_n)|H,P_n}$, the inequality reads:

$$\rho[\mathbb{P}_{(P,\xi)|H}\{\mathsf{A}_{\mathrm{conv}}\}] \geq \rho\left[\hat{\mathbb{P}}_{(P,\xi)|H}\{\mathsf{A}\}\right]$$
$$+ \frac{D_{\mathrm{KL}}(\rho \parallel \mathbb{P}_H) + \log\left(\frac{1}{\varepsilon}\right)}{\lambda}.$$

This is intuitive: For larger $N$, we have more confidence in our estimate $\hat{\mathbb{P}}_{(P,\xi)|H}\{\mathsf{A}\}$, so we can choose a larger $\lambda$ which dampens the last term and tightens the lower bound for $\mathbb{P}_{(P,\xi)|H}\{\mathsf{A}_{\mathrm{conv}}\}$.

# 8. Experiments

In this section, we conduct two experiments: The strongly convex and smooth problem of minimizing quadratic functions with varying strong convexity, varying smoothness, and varying right-hand side, and the non-smooth non-convex problem of training a neural network on different data sets. The code to reproduce the results can be found at https://github.com/MichiSucker/COLA_2024.

## 8.1. Quadratic Problems

First, we train the algorithm $\mathcal{A}$ to solve quadratic problems. Thus, each optimization problem $\ell(\cdot, p)$ is of the form

$$\min_{z \in \mathbb{R}^d} \frac{1}{2}\|Az - b\|^2, \quad A \in \mathbb{R}^{d \times d}, \ b \in \mathbb{R}^d,$$

such that the parameters are given by $p = (A, b) \in \mathbb{R}^{d^2+d} =: \mathscr{P}$, and the optimization variable is $z \in \mathbb{R}^d$, $d = 200$. The strong-convexity and smoothness constants of $\ell$ are sampled randomly in the intervals $[m_-, m_+], [L_-, L_+] \subset (0, +\infty)$, and we define the matrix $A_j$, $j = 1, ..., N$, as *diagonal matrix* with entries $a_{ii}^j = \sqrt{m_j} + i(\sqrt{L_j} - \sqrt{m_j})/d$, $i = 1, ..., d$. In principle, this is a severe restriction. However, we do not use this knowledge explicitly in the design of our algorithm $\mathcal{A}$,

that is, if the algorithm "finds" this structure during learning by itself, it can leverage on it. Like this, the given class of functions is $L_+$-smooth and $m_-$-strongly convex, such that we use *heavy-ball with friction* (HBF) (Polyak, 1964) as worst-case optimal baseline. Its update is given by $z^{(t+1)} = z^{(t)} - \beta_1 \nabla f(z^{(t)}) + \beta_2 \left(z^{(t)} - z^{(t-1)}\right)$, where the optimal worst-case convergence rate is attained for $\beta_1 = \left(\frac{2}{\sqrt{L_+}+\sqrt{\mu_-}}\right)^2, \beta_2 = \left(\frac{\sqrt{L_+}-\sqrt{\mu_-}}{\sqrt{L_+}+\sqrt{\mu_-}}\right)^2$ (Nesterov, 2018). Similarly, the learned algorithm $\mathcal{A}$ performs an update of the form $z^{(t+1)} = z^{(t)} + \beta^{(t)}d^{(t)}$, where $\beta^{(t)}$ and $d^{(t)}$ are predicted by separate blocks of a neural network. Here, we stress that the update is not constrained in any way. For more details on the architecture we refer to Appendix G. Since the functions are smooth and strongly convex, we only have to check the sufficient-descent condition and the relative-error condition. Obviously, in practice it is impossible to check them for all $t \in \mathbb{N}_0$. Thus, we restrict to $t_{\mathrm{train}} = 500$ iterations. Then, given a measurable selection $v(z, p) \in \partial_1 \ell(z, p)$, the relative-error condition is trivially satisfied with $b := \max_{t \leq t_{\mathrm{train}}}\{\|v(z^{(t)}, p)\|\}/\min_{t \leq t_{\mathrm{train}}}\|z^{(t)} - z^{(t-1)}\|$, such that we only have to check the sufficient-descent condition during training. Finally, we consider $\xi$ to be converged, if the loss is smaller than $10^{-16}$. For more details we refer again to Appendix G. The results are shown in Figure 1: The left plot shows the distance to the minimizer $z^*$ over the iterations, where HBF is shown in blue and the learned algorithm in pink, and we can see that the learned algorithm is clearly superior. The right plot shows the estimated probabilities $\mathbb{P}_{(P,\xi)|H}\{\mathsf{A}\}$ (yellow dashed line), $\mathbb{P}_{(P,\xi)|H}\{\mathsf{A}_{\mathrm{conv}}\}$ (purple dashed line), and the PAC-bound (orange dotted line) on 250 test sets of size $N = 250$. We can see that the PAC-bound is quite tight for $\mathbb{P}_{(P,\xi)|H}\{\mathsf{A}\}$, while there is a substantial gap $\mathbb{P}_{(P,\xi)|H}\{\mathsf{A}_{\mathrm{conv}} \setminus \mathsf{A}\}$. Nevertheless, it *guarantees* convergence of the learned algorithm in about 75% of the test problems.

## 8.2. Training a Neural Network

As second experiment, we train the algorithm $\mathcal{A}$ to train a neural network $\mathtt{N}$ on a regression problem. Thus, the algorithm $\mathcal{A}$ predicts parameters $\beta \in \mathbb{R}^d$, such that $\mathtt{N}(\beta, \cdot)$ estimates a function $g : \mathbb{R} \to \mathbb{R}$ from noisy observations $y_{i,j} = g_i(x_j) + \varepsilon_{i,j}$, $i = 1, ..., N$, $j = 1, ..., K$ ($K = 50$), with $\varepsilon_{i,j} \overset{iid}{\sim} \mathcal{N}(0, 1)$. Here, we use the mean square error as loss for the neural network, and for $\mathtt{N}$ we use a fully-connected two layer neural network with ReLU-activation functions. Then, by using the data sets as parameters, that is, $\mathscr{P} = \mathbb{R}^{K \times 2}$ and $p_i = \{(x_{i,j}, y_{i,j})\}_{j=1}^{K}$, the loss functions for the algorithm are given by $\ell(\beta, p_i) := \frac{1}{K}\sum_{j=1}^{K}(\mathtt{N}(\beta, x_{i,j}) - y_{i,j})^2$, which are non-smooth and non-convex in $\beta$. Here, the input $x$ is transformed into the vector $(x, x^2, ..., x^5)$, such that the parameters $\beta \in \mathbb{R}^d$ are given by the weights $A_1 \in \mathbb{R}^{50 \times 5}, A_2 \in \mathbb{R}^{1 \times 50}$ and

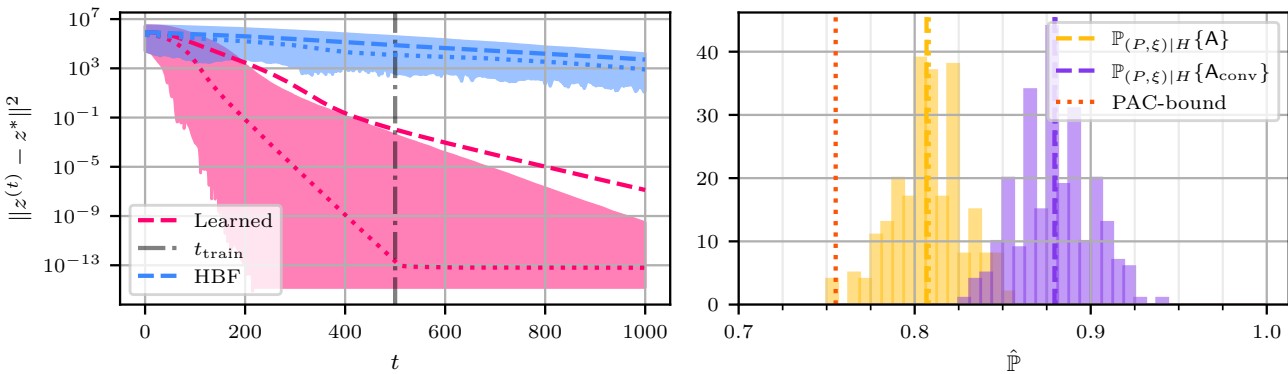

*Figure 1.* Quadratic problems: The left figure shows the distance to the minimizer over the iterations, where *heavy-ball with friction* (HBF) is shown in blue and the learned algorithm in pink. The mean and median are shown as dashed and dotted lines, respectively, while the shaded region represents 95% of the test data. One can see that the learned algorithm converges way faster than HBF. The right plot shows the estimates (dashed lines) for $\mathbb{P}_{(P,\xi)|H}\{A\}$ (orange), $\mathbb{P}_{(P,\xi)|H}\{A_{\mathrm{conv}}\}$ (purple), and the PAC-bound (dark orange). One can see that the predicted chain of inequalities $1 - \Phi^{-1}(...) \leq \mathbb{P}_{(P,\xi)|H}\{A\} \leq \mathbb{P}_{(P,\xi)|H}\{A_{\mathrm{conv}}\}$ does hold true.

biases $b_1 \in \mathbb{R}^{50}, b_2 \in \mathbb{R}$ of the two fully-connected layers. Thus, the optimization space is of dimension $p = 351$. For the functions $g_i$ we use polynomials of degree $d = 5$, where we sample the coefficients $(c_{i,0}, ..., c_{i,5})$ uniformly in $[-5, 5]$. Similarly, we sample the points $\{x_{i,j}\}_{j=1}^{K}$ uniformly in $[-2, 2]$. As baseline we use Adam (Kingma & Ba, 2015) as it is implemented in PyTorch (Paszke et al., 2019), and we tune its step-size with a simple grid search over 100 values in $[10^{-4}, 10^{-2}]$, such that its performance is best for the given $t_{\mathrm{train}} = 250$ iterations. This yields the value $\kappa = 0.008$. Note that we use Adam in the "full-batch setting" here, while, originally, it was introduced for the stochastic case. On the other hand, the learned algorithm performs the update $z^{(t+1)} = z^{(t)} + d^{(t)}/\sqrt{t}$, where $d^{(t)}$ is predicted by a neural network. Again, we stress that $d^{(t)}$ is not constrained in any way. For more details on the architecture, we refer to Appendix I. As we cannot access the critical points directly, we approximate them by running gradient descent for $5 \cdot 10^4$ iterations with a step-size of $1 \cdot 10^{-6}$, starting for each problem and algorithm from the last iterate ($t = 500$). Similarly, we cannot estimate the convergence probability in this case, only the probability for the event A. The results of this experiment are shown in Figure 2: The left plot shows the distance to the critical point and the plot in the middle shows the loss. Here, Adam is shown in blue, while the learned algorithm is shown in pink. Finally, the right plot shows the estimate for $\mathbb{P}_{(P,\xi)|H}\{A\}$ and the predicted PAC-bound. We can see that the learned algorithm does indeed seem to converge to a critical point and it minimizes the loss faster than Adam. Further, the PAC-bound is quite tight, and it *guarantees* that the learned algorithm will converge in about 92% of the problems.

## 9. Conclusion

We presented a novel method for deducing convergence of generic learned algorithms that exhibit a Markovian structure with high probability. To showcase the idea, we derived a new convergence result for learned optimization algorithms on (possibly) non-smooth non-convex loss-functions based on generalization. This was based on the fundamental insight that, contrary to traditional optimization, in learning-to-optimize we can actually *observe* the algorithm during training. While the approach is theoretically sound, practically it has at least four drawbacks, on which we shortly want to comment: First, and foremost, one simply cannot observe the *whole* trajectory in practice. Thus, one can only obtain an approximation to this result, that is, whether the used conditions do hold up to a certain number of iterations. Nevertheless, by using sufficiently many iterations, one can guarantee that the algorithm gets sufficiently close to a critical point. Second, instead of verifying the conditions used here, one could alternatively try to observe the final result directly, for example by looking at the norm of the gradient. However, when checking the proposed conditions one is guaranteed to get arbitrary close to a critical point, while, in the other case, one could end up with a small gradient norm that is arbitrary far away from a critical point. Especially, this applies in the non-smooth setting, where the subdifferential does not necessarily tell anything about the distance to a critical point[3], or to applications where one simply cannot access critical points during training. Third, for now, training the algorithm in such a way that it actually does satisfy the proposed properties on a majority of problems

---

[3]For example, consider $f(z) = |z|$.

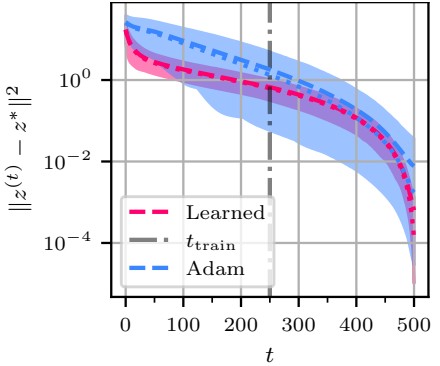 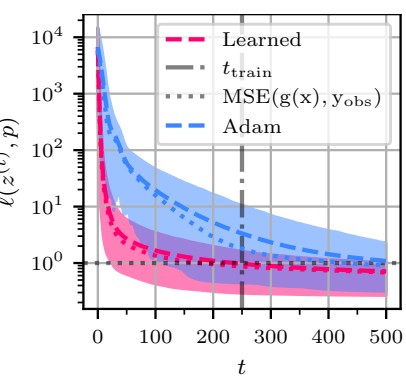 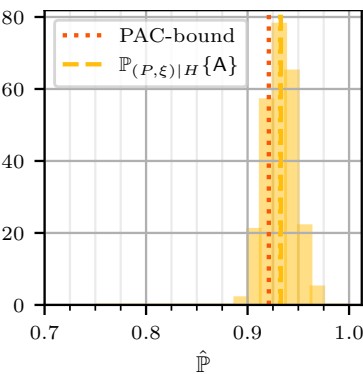

*Figure 2.* Training a neural network: The left figure shows the distance to the estimated critical point and the figure in the middle shows the loss. Adam is shown in blue and the learned algorithm in pink. The mean and median are shown as dashed and dotted lines, respectively, while the shaded region represents 95% of the test data. We see that the learned algorithm minimizes the loss faster than Adam, and seems to converge to a critical point. The right plot shows the estimate for $\mathbb{P}_{(P,\xi)|H}\{A\}$ (orange dashed line) and the PAC-bound (dark orange).

is quite difficult and time-consuming. Lastly, due to the sufficient-descent condition, Theorem 6.5 is not well-suited for stochastic optimization. Nevertheless, Theorem 6.3 and the proposed approach can directly be transferred to the stochastic setting, which we leave for future work.

## Acknowledgements

M. Sucker and P. Ochs acknowledge funding by the German Research Foundation under Germany's Excellence Strategy – EXC number 2064/1 – 390727645.

## Impact Statement

We do not see any negative impact of our presented work.

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

## A. Missing Definitions

The following definitions can be found in the book of Rockafellar & Wets (1998). A function $f : \mathbb{R}^d \to \mathbb{R} \cup \{\pm\infty\}$ is called *proper*, if $f(x) < +\infty$ for at least one point $x \in \mathbb{R}^d$ and $f(x) > -\infty$ for all $x \in \mathbb{R}^d$. In this case, the *effective domain* of $f$ is the set

$$\operatorname{dom} f := \{x \in \mathbb{R}^d \ : \ f(x) < +\infty\} \,.$$

Similarly, for a set-valued mapping $T : \mathcal{X} \rightrightarrows \mathcal{Y}$ the *graph* is defined as

$$\operatorname{gph} T := \{(x, y) \in \mathcal{X} \times \mathcal{Y} \ : \ y \in T(x)\} \,,$$

while its *domain* is defined as

$$\operatorname{dom} T := \{x \in \mathcal{X} \ : \ T(x) \neq \emptyset\} \,.$$

The *outer limit* of a set-valued map $T : \mathbb{R}^k \rightrightarrows \mathbb{R}^l$ is defined as:

$$\limsup_{x \to \bar{x}} T(x) := \left\{ y \mid \exists x^{(t)} \to \bar{x}, \ \exists y^{(t)} \to y \text{ with } y^{(t)} \in T(x^{(t)}) \right\} \,.$$

Based on this, $T$ is said to be *outer semi-continuous* at $\bar{x}$, if

$$\limsup_{x \to \bar{x}} T(x) \subset T(\bar{x}) \,.$$

**Definition A.1.** Consider a function $f : \mathbb{R}^d \to \mathbb{R} \cup \{\pm\infty\}$ and a point $\bar{x} \in \operatorname{dom} f$. For a vector $v \in \mathbb{R}^d$, one says that

(i) $v$ is a *regular subgradient* of $f$ at $\bar{x}$, if

$$f(x) \geq f(\bar{x}) + \langle v, x - \bar{x} \rangle + o\left(\|x - \bar{x}\|\right) \,.$$

The set of regular subgradients of $f$ at $\bar{x}$, denoted by $\hat{\partial} f(\bar{x})$, is called the *regular subdifferential* of $f$ at $\bar{x}$.

(ii) $v$ is a (general) *subgradient* of $f$ at $\bar{x}$, if there are sequences $x^{(t)} \to \bar{x}$ and $v^{(t)} \to v$ with $f(x^{(t)}) \to f(\bar{x})$ and $v^{(t)} \in \hat{\partial} f(x^{(t)})$. The set of subgradients of $f$ at $\bar{x}$, denoted by $\partial f(\bar{x})$, is called the (limiting) *subdifferential* of $f$ at $\bar{x}$.

Finally, the following definition can be found in Attouch et al. (2013, Definition 2.4, p.7).

**Definition A.2.**    a) The function $f : \mathbb{R}^d \to \mathbb{R} \cup \{+\infty\}$ is said to have the *Kurdyka-Łojasiewicz property* at $\bar{x} \in \operatorname{dom} \partial f$, if there exist $\eta \in (0, +\infty]$, a neighborhood $U$ of $\bar{x}$, and a continuous concave function $\varphi : [0, \eta) \to [0, \infty)$, such that

   (i) $\varphi(0) = 0$,
   (ii) $\varphi$ is $C^1$ on $(0, \eta)$,
   (iii) for all $s \in (0, \eta)$, $\varphi'(s) > 0$,
   (iv) for all $x$ in $U \cap \{f(\bar{x}) < f < f(\bar{x}) + \eta\}$, the Kurdyka-Łojasiewicz inequality holds

$$\varphi'\left(f(x) - f(\bar{x})\right) \cdot \operatorname{dist}(0, \partial f(x)) \geq 1 \,.$$

   b) Proper lower semi-continuous functions which satisfy the Kurdyka-Łojasiewicz property at each point of $\operatorname{dom} \partial f$ are called *Kurdyka-Łojasiewicz functions*.

## B. Counterexamples

**Example B.1** (Violation of boundedness assumption)**.** Starting from $z^{(1)} := 1$, define the sequence for $2 \leq t \in \mathbb{N}$ by $z^{(t)} := z^{(t-1)} + \frac{1}{t}$, and consider the positive and convex function $f(z) := \exp(-z)$. We show that the sequence $(z^{(t)})_{t \in \mathbb{N}}$ does satisfy the sufficient-descent condition for $f$: By definition, we have the recursive formula $f(z^{(t+1)}) = f(z^{(t)}) \exp\left(-\frac{1}{t+1}\right)$, which allows for rewriting the sufficient-descent condition as:

$$\frac{a}{(t+1)^2} \leq \left(1 - \exp\left(-\frac{1}{t+1}\right)\right) f(z^{(t)}) \,.$$

Then, we have to find $a > 0$ satisfying this inequality for all $t \in \mathbb{N}$. For $t = 1$, the right-hand side is greater than $\frac{1}{9}$, such that we can choose any $a \in (0, \frac{4}{9}]$ (rough estimate). Thus, take $a \in (0, \frac{4}{9}]$, such that $\tilde{a} := a \cdot 2e \in (0, \frac{4}{9}]$, and proceed by induction (note that $\tilde{a}$ satisfies the stated condition for $t = 1$). Assuming that the inequality holds true for up to time $t$, we get by the induction hypothesis:

$$\frac{\tilde{a}}{(t+2)^2} \le \left(\frac{t+1}{t+2}\right)^2 \left(1 - \exp\left(-\frac{1}{t+1}\right)\right) f(z^{(t)})$$

By inserting a trivial 1 three-times, the right-hand side can be written as:

$$\left(\frac{t+1}{t+2}\right)^2 \cdot \exp\left(\frac{1}{t+2}\right) \cdot \frac{\exp\left(\frac{1}{t+1}\right) - 1}{\exp\left(\frac{1}{t+2}\right) - 1} \cdot \left(1 - \exp\left(-\frac{1}{t+2}\right)\right) f(z^{(t+1)}).$$

Here, the first term is bounded by 1, the second by $e$, and the third by 2. Hence, dividing both sides by $2e$, we get:

$$\frac{a}{(t+2)^2} \le \left(1 - \exp\left(-\frac{1}{t+2}\right)\right) f(z^{(t+1)}),$$

such that $(z^{(t)})_{t \in \mathbb{N}}$ satisfies the sufficient-descent condition for $f$. Nevertheless, we have $z^{(t)} = z^{(t-1)} + \frac{1}{t} = ... = \sum_{k=1}^{t} \frac{1}{k}$, such that $|z^{(t)}| \overset{t \to \infty}{\to} \infty$, that is, the sequence is unbounded and does not converge.

**Example B.2** (Violation of relative-error condition). Consider the smooth and strongly convex function $f(z_1, z_2) := \frac{1}{2}z_1^2 + \frac{1}{2}z_2^2$, and define the sequence $((z^{(t)}, z_2^{(t)}))_{t \in \mathbb{N}_0} \subset \mathbb{R}^2$ through

$$(z_1^{(t+1)}, z_2^{(t+1)}) := (z_1^{(t)} - 0.1z_1^{(t)}, z_2^{(t)}).$$

Then we have $\|(z_1^{(t+1)}, z_2^{(t+1)}) - (z_1^{(t)}, z_2^{(t)})\|^2 = (0.1z_1^{(t)})^2$, and it holds:

$$\begin{aligned}
f(z_1^{(t+1)}, z_2^{(t+1)}) &= \frac{1}{2}\left(z_1^{(t)} - 0.1z_1^{(t)}\right)^2 + \frac{1}{2}\left(z_2^{(t)}\right)^2 \\
&= f(z_1^{(t)}, z_2^{(t)}) - 0.1\left(z_1^{(t)}\right)^2 + \frac{1}{2}\left(0.1z_1^{(t)}\right)^2 \\
&= f(z_1^{(t)}, z_2^{(t)}) - 9.5\|(z_1^{(t+1)}, z_2^{(t+1)}) - (z_1^{(t)}, z_2^{(t)})\|^2,
\end{aligned}$$

such that $(z_1^{(t+1)}, z_2^{(t+1)})$ satisfies the sufficient-descent condition. However, for $z_2^{(0)} \ne 0$, the sequence converges to $(0, z_2^{(0)})$, which is not a critical point of $f$.

## C. Proof of Lemma 7.1

**Lemma C.1.** *Suppose Assumption 6.7 holds. Then, $\mathsf{A}_{\mathrm{crit}}$ is closed.*

*Proof.* Take $(p^{(t)}, z^{(t)})_{t \in \mathbb{N}} \subset \mathsf{A}_{\mathrm{crit}}$ with $(p^{(t)}, z^{(t)}) \to (\bar{p}, \bar{z}) \in \mathscr{P} \times \mathfrak{Z}$. We need to show that $(\bar{p}, \bar{z}) \in \mathsf{A}_{\mathrm{crit}}$. Since $(z, p) \mapsto \partial_1 \ell(z, p)$ is outer semi-continuous, we have:

$$\limsup_{(z,p) \to (\bar{z}, \bar{p})} \partial_1 \ell(z, p) \subset \partial_1 \ell(\bar{z}, \bar{p}).$$

By definition of the outer limit, this is the same as:

$$\left\{ u \in \mathfrak{Z} \mid \exists (z^{(t)}, p^{(t)}) \to (\bar{z}, \bar{p}), \exists u^{(t)} \to u \text{ with } u^{(t)} \in \partial_1 \ell(z^{(t)}, p^{(t)}) \right\} \subset \partial_1 \ell(\bar{z}, \bar{p}).$$

In particular, we have that $(z^{(t)}, p^{(t)})_{t \in \mathbb{N}} \to (\bar{z}, \bar{p})$, and it holds $0 \in \partial_1 \ell(z^{(t)}, p^{(t)})$ for all $t \in \mathbb{N}$. Thus, setting $u^{(t)} := 0$ for all $t \in \mathbb{N}$ and $u := 0$, we conclude that $0 \in \partial_1 \ell(\bar{z}, \bar{p})$. Hence, $(\bar{p}, \bar{z}) \in \mathsf{A}_{\mathrm{crit}}$, and $\mathsf{A}_{\mathrm{crit}}$ is closed. $\square$

Now, we can prove Lemma 7.1:

*Proof.* To show measurability of $\mathsf{A}_{\mathrm{conv}}$, we adopt the notation of the *limes inferior* for sets from probability theory: If $d$ is a metric on $\mathscr{P} \times \mathfrak{X}$ and $\varepsilon > 0$, define the set

$$\{\mathsf{B}_\varepsilon(p, z) \text{ ult.}\} := \left\{ (p', z^{(t)}) \in \mathsf{B}_\varepsilon(p, z) \text{ ult.} \right\} := \bigcup_{n \in \mathbb{N}_0} \bigcap_{t \geq n} \left\{ (p', z^{(t)}) \in \mathsf{B}_\varepsilon(p, z) \right\} .$$

Here, $\left\{ (p', z^{(t)}) \in \mathsf{B}_\varepsilon(p, z) \right\}$ is a short-hand notation for $\{(p', (z^{(t)})_{t \in \mathbb{N}_0}) \in \mathscr{P} \times \mathfrak{X}^{\mathbb{N}_0} : (p', z^{(t)}) \in \mathsf{B}_\varepsilon(p, z)\}$. Thus, $\{\mathsf{B}_\varepsilon(p, z) \text{ ult.}\}$ is the (parametric) set of all sequences in $\mathfrak{X}$ that *ultimately* lie in the ball with radius $\varepsilon$ around $(p, z)$. Note that $\{\mathsf{B}_\varepsilon(p, z) \text{ ult.}\}$ is measurable w.r.t. to the product $\sigma$-algebra on $\mathscr{P} \times \mathfrak{X}^{\mathbb{N}_0}$, since it is the countable union/intersection of measurable sets, where $\{(p', z^{(t)}) \in \mathsf{B}_\varepsilon(p, z)\}$ is measurable, since it can be written as $\{d((p', z^{(t)}), (p, z)) < \varepsilon\} = (g \circ (id, \pi_t))^{-1}[0, \varepsilon)$. Here, $id$ is the identity on $\mathscr{P}$, and $g(p', z') := d((p', z'), (p, z))$ is continuous.

Since the proof does not get more complicated by considering general Polish space $\mathscr{P}, \mathfrak{X}$ instead of $\mathbb{R}^q$ and $\mathbb{R}^d$, we prove the result in this more general setting. For this, denote the the complete metric on $\mathscr{P}$ by $d_{\mathscr{P}}$, and the one on $\mathfrak{X}$ by $d_{\mathfrak{X}}$. Then we have that $d_{\mathscr{P} \times \mathfrak{X}} := d_{\mathscr{P}} + d_{\mathfrak{X}}$ is a metric on $\mathscr{P} \times \mathfrak{X}$ that metrizes the product-topology, that is, it yields the same $\sigma$-algebra. Similarly, denote the countable dense subset in $\mathscr{P}$ by $\mathcal{P}$, and the one in $\mathfrak{X}$ by $\mathcal{Z}$. Then we have that $\mathcal{D} := \mathcal{P} \times \mathcal{Z}$ is a countable and dense subset of $\mathscr{P} \times \mathfrak{X}$.

If $\mathsf{A}_{\mathrm{crit}}$ is empty, we get that $\mathsf{A}_{\mathrm{conv}} = \emptyset$, which is measurable. Hence, w.l.o.g. assume that $\mathsf{A}_{\mathrm{crit}} \neq \emptyset$. We claim that:

$$\mathsf{A}_{\mathrm{conv}} = \mathsf{C} := \bigcap_{k \in \mathbb{N}} \bigcup_{\substack{(p, z) \in \mathcal{D} \\ \mathsf{A}_{\mathrm{crit}} \cap \mathsf{B}_{1/k}(p, z) \neq \emptyset}} \left\{ \mathsf{B}_{1/k}(p, z) \text{ ult.} \right\} .$$

If this equality holds, $\mathsf{A}_{\mathrm{conv}}$ is measurable as a countable intersection/union of measurable sets. Thus, it remains to show the equality $\mathsf{A}_{\mathrm{conv}} = \mathsf{C}$, which we do by showing both inclusions. Therefore, first, take $(p, (z^{(t)})_{t \in \mathbb{N}_0}) \in \mathsf{A}_{\mathrm{conv}}$. Then there exists $z^* \in \mathfrak{X}$, such that $(p, z^*) \in \mathsf{A}_{\mathrm{crit}}$ and $\lim_{t \to \infty} d_{\mathfrak{X}}(z^{(t)}, z^*) = 0$. Hence, for any $k \in \mathbb{N}$, there exists $t_k \in \mathbb{N}$, such that $z^{(t)} \in \mathsf{B}_{1/3k}(z^*)$ for all $t \geq t_k$. Now, take $(p_k, z_k) \in \mathcal{D}$, such that $p_k \in \mathsf{B}_{1/3k}(p)$ and $z_k \in \mathsf{B}_{1/3k}(z^*)$, which exists, since $\mathcal{D}$ is dense. Then, for all $t \geq t_k$ we have:

$$d_{\mathscr{P} \times \mathfrak{X}}((p, z^{(t)}), (p_k, z_k)) = d_{\mathscr{P}}(p, p_k) + d_{\mathfrak{X}}(z^{(t)}, z_k) \leq d_{\mathscr{P}}(p, p_k) + d_{\mathfrak{X}}(z^{(t)}, z^*) + d_{\mathfrak{X}}(z^*, z_k) < \frac{1}{k} ,$$

that is, $(p, (z^{(t)})_{t \in \mathbb{N}_0}) \in \{\mathsf{B}_{1/k}(p_k, z_k) \text{ ult.}\}$. Further, we have:

$$d_{\mathscr{P} \times \mathfrak{X}}((p, z^*), (p_k, z_k)) < \frac{2}{3k} < \frac{1}{k} .$$

Hence, $(p_k, z_k) \in \mathcal{D}$ with $\mathsf{A}_{\mathrm{crit}} \cap \mathsf{B}_{1/k}(p_k, z_k) \neq \emptyset$. Since such a tuple $(p_k, z_k) \in \mathcal{D}$ can be found for any $k \in \mathbb{N}$, we get:

$$(p, (z^{(t)})_{t \in \mathbb{N}_0}) \in \bigcup_{\substack{(p', z') \in \mathcal{D} \\ \mathsf{A}_{\mathrm{crit}} \cap \mathsf{B}_{1/k}(p', z') \neq \emptyset}} \{\mathsf{B}_{1/k}(p', z') \text{ ult.}\}, \quad \forall k \in \mathbb{N} .$$

Then, however, this implies $(p, (z^{(t)})_{t \in \mathbb{N}_0}) \in \mathsf{C}$, which shows the inclusion $\mathsf{A}_{\mathrm{conv}} \subset \mathsf{C}$. Now, conversely, let $(p, (z^{(t)})_{t \in \mathbb{N}_0}) \in \mathsf{C}$. Then, for every $k \in \mathbb{N}$ there exists $(p_k, z_k) \in \mathcal{D}$ with $\mathsf{A}_{\mathrm{crit}} \cap \mathsf{B}_{1/k}(p_k, z_k) \neq \emptyset$, and a $t_k \in \mathbb{N}$, such that

$$(p, z^{(t)}) \in \mathsf{B}_{1/k}(p_k, z_k), \quad \forall t \geq t_k .$$

The resulting sequence of midpoints $(p_k, z_k)_{k \in \mathbb{N}}$ is Cauchy in $\mathscr{P} \times \mathfrak{X}$, because: For $k, l \in \mathbb{N}$, we have that $(p, z^{(t)}) \in \mathsf{B}_{1/k}(p_k, z_k)$ for all $t \geq t_k$, and $(p, z^{(t)}) \in \mathsf{B}_{1/l}(p_l, z_l)$ for all $t \geq t_l$. Thus, for $t \geq T := \max\{t_k, t_l\}$, we get $(p, z^{(t)}) \in \mathsf{B}_{1/k}(p_k, z_k) \cap \mathsf{B}_{1/l}(p_l, z_l)$, which allows for the following bound:

$$\begin{aligned} d_{\mathscr{P} \times \mathfrak{X}}((p_k, z_k), (p_l, z_l)) &\leq d_{\mathscr{P} \times \mathfrak{X}}((p_k, z_k), (p, z^{(t_k)})) + d_{\mathscr{P} \times \mathfrak{X}}((p, z^{(t_k)}), (p, z^{(T)})) \\ &\quad + d_{\mathscr{P} \times \mathfrak{X}}((p, z^{(T)}), (p, z^{(t_l)})) + d_{\mathscr{P} \times \mathfrak{X}}((p, z^{(t_l)}), (p_l, z_l)) \\ &\leq \frac{1}{k} + \frac{2}{k} + \frac{2}{l} + \frac{1}{l} \leq \frac{3}{k} + \frac{3}{l} \overset{k, l \to \infty}{\to} 0 . \end{aligned}$$

Hence, by completeness of $\mathscr{P} \times \mathfrak{X}$, the sequence $(p_k, z_k)_{k \in \mathbb{N}}$ has a limit $(p^*, z^*)$ in $\mathscr{P} \times \mathfrak{X}$. First, we show that $p^* = p$: Since $(p, z^{(t_k)}) \in \mathsf{B}_{1/k}(p_k, z_k)$ for all $k \in \mathbb{N}$, we have by continuity of the metric:

$$d_{\mathscr{P}}(p, p^*) = \lim_{k \to \infty} d_{\mathscr{P}}(p, p_k) \leq \lim_{k \to \infty} d_{\mathscr{P} \times \mathfrak{X}}((p, z^{(t_k)}), (p_k, z_k)) \leq \lim_{k \to \infty} \frac{1}{k} = 0 \,.$$

Thus, actually, $(p_k, z_k) \to (p, z^*)$. Second, we show that $(p, z^*) \in \mathsf{A}_{\mathrm{crit}}$, that is, $z^* \in \mathsf{A}_{\mathrm{crit},p}$: Assume the contrary, that is, $(p, z^*) \in \mathsf{A}_{\mathrm{crit}}^c$. By Lemma C.1, the set $\mathsf{A}_{\mathrm{crit}}$ is closed. Thus, its complement $\mathsf{A}_{\mathrm{crit}}^c$ is open, and there exists $\varepsilon > 0$ with $\mathsf{B}_\varepsilon(p, z^*) \subset \mathsf{A}_{\mathrm{crit}}^c$, that is, $\mathsf{B}_\varepsilon(p, z^*) \cap \mathsf{A}_{\mathrm{crit}} = \emptyset$. Since $(p_k, z_k) \to (p, z^*)$, there exists $N \in \mathbb{N}$, such that $d_{\mathscr{P} \times \mathfrak{X}}((p_k, z_k), (p, z^*)) < \frac{\varepsilon}{3}$ for all $k \geq N$. Then, however, taking $k \geq N$ with $\frac{1}{k} < \frac{\varepsilon}{3}$, we conclude that

$$\mathsf{B}_{1/k}(p_k, z_k) \cap \mathsf{A}_{\mathrm{crit}} = \emptyset \,.$$

By definition of the sequence $(p_k, z_k)_{k \in \mathbb{N}}$, this is a contradiction. Hence, we have $(p, z^*) \in \mathsf{A}_{\mathrm{crit}}$, and it remains to show that also the sequence $(z^{(t)})_{t \in \mathbb{N}_0}$ converges to $z^*$. For this, assume the contrary again. Then there exists an $\varepsilon > 0$ with the property that for all $T \in \mathbb{N}$, one can find a $\tilde{t} \geq T$, such that $d_{\mathfrak{X}}(z^{(\tilde{t})}, z^*) \geq \varepsilon$. Now, choose $k \in \mathbb{N}$ large enough, such that $d_{\mathfrak{X}}(z_k, z^*) \leq \frac{\varepsilon}{3}$ and $\frac{1}{k} < \frac{\varepsilon}{3}$. Then, since $(p, z^{(t)}) \in \mathsf{B}_{1/k}(p_k, z_k)$ for all $t \geq t_k$, we have:

$$d_{\mathfrak{X}}(z^{(t)}, z^*) \leq d_{\mathfrak{X}}(z^{(t)}, z_k) + d_{\mathfrak{X}}(z_k, z^*) \leq \frac{2\varepsilon}{3} < \varepsilon, \quad \forall t \geq t_k \,.$$

Again, this is a contradiction and such an $\varepsilon > 0$ cannot exists. Thus, $(z^{(t)})_{t \in \mathbb{N}_0}$ converges to $z^* \in \mathsf{A}_{\mathrm{crit},p}$, and we have $(p, (z^{(t)})_{t \in \mathbb{N}_0}) \in \mathsf{A}_{\mathrm{conv}}$, which concludes the proof. $\qquad \square$

## D. Proof of Lemma 7.2

*Proof.* Since $\mathbb{Q}$ is dense in $\mathbb{R}$, we can restrict to $a \in (0, \infty) \cap \mathbb{Q} =: \mathbb{Q}_+$. Then $\mathsf{A}_{\mathrm{desc}}$ can be written as

$$\left( \bigcup_{a \in \mathbb{Q}_+} \bigcap_{t \in \mathbb{N}_0} \mathsf{A}_{a,t} \right) \cap \left( \bigcap_{t \in \mathbb{N}_0} \left\{ \ell(z^{(t)}, p) < \infty \right\} \right) ,$$

where $\mathsf{A}_{a,t}$ is given by:

$$\left\{ \ell(z^{(t+1)}, p) + a \| z^{(t+1)} - z^{(t)} \|^2 \leq \ell(z^{(t)}, p) \right\} \,.$$

Since $\sigma$-algebras are stable under countable unions/intersection, it suffices to show that the sets $\{ \ell(z^{(t)}, p) < \infty \}$ and $\mathsf{A}_{a,t}$ are measurable. Here, the set $\{ \ell(z^{(t)}, p) < \infty \}$ can be written as:

$$\left\{ \ell(z^{(t)}, p) < \infty \right\} = (\ell \circ \Phi \circ (id, \pi_t))^{-1} [0, \infty) ,$$

where $\Phi : \mathscr{P} \times \mathfrak{X} \to \mathfrak{X} \times \mathscr{P}$ just interchanges the coordinates (which is measurable), and $id$ is the identity on $\mathscr{P}$. Since $[0, \infty)$ is a measurable set and $\ell$ is assumed to be measurable, we have that $\{ \ell(z^{(t)}, p) < \infty \}$ is measurable for each $t \in \mathbb{N}_0$. To show that $\mathsf{A}_{a,t}$ is measurable, we define the function $g_a : (\mathrm{dom}\,\ell)^2 \to \mathbb{R}$ through $((z_1, p_1), (z_2, p_2)) \mapsto \ell(z_2, p_2) - \ell(z_1, p_1) + a \| z_2 - z_1 \|^2$. Then, $g_a$ is measurable and $\mathsf{A}_{a,t}$ can be written as:

$$\begin{aligned}
\mathsf{A}_{a,t} &= \left\{ g_a(z^{(t)}, p, z^{(t+1)}, p) \leq 0 \right\} \\
&= \left\{ (g_a \circ (\pi_t, id, \pi_{t+1}, id) \circ \iota) \left( p, (z^{(t)})_{t \in \mathbb{N}_0} \right) \leq 0 \right\} \\
&= (g_a \circ (\pi_t, id, \pi_{t+1}, id) \circ \iota))^{-1} (-\infty, 0] ,
\end{aligned}$$

where $\iota : \mathscr{P} \times \mathfrak{X}^{\mathbb{N}_0} \to (\mathfrak{X}^{\mathbb{N}_0} \times \mathscr{P})^2$ is the diagonal inclusion $(p, z) \mapsto ((z, p), (z, p))$, which is measurable w.r.t. to the product-$\sigma$-algebra on $(\mathfrak{X}^{\mathbb{N}_0} \times \mathscr{P})^2$, since $\iota^{-1}(\mathsf{B}_1 \times \mathsf{B}_2) = \mathsf{B}_1 \cap \mathsf{B}_2$. Thus, the set $\mathsf{A}_{a,t}$ is measurable, which concludes the proof. $\qquad \square$

## E. Existence of Measurable Selection and Proof of Lemma 7.3

**Definition E.1.** A set-valued mapping $T : \mathcal{X} \rightrightarrows \mathbb{R}^d$ is *measurable*, if for every open set $\mathsf{O} \subset \mathbb{R}^d$ the set $T^{-1}(\mathsf{O}) \subset \mathcal{X}$ is measurable. In particular, $\mathrm{dom}\, T$ has to be measurable.

**Lemma E.2.** *Suppose Assumption 6.7 holds. Then $(z, p) \mapsto \partial_1 \ell(z, p)$ is closed-valued and measurable.*

*Proof.* Since $\partial_1 \ell(z, p)$ is the subdifferential of $\ell(\cdot, p)$ at $z$, by Rockafellar & Wets (1998, Theorem 8.6, p.302) we have that the set $\partial_1 \ell(z, p)$ is closed for every $p \in \mathcal{P}$ and every $z \in \mathrm{dom}\, \ell(\cdot, p)$. Hence, we have that $\partial_1 \ell(z, p)$ is closed for every $(z, p) \in \mathrm{dom}\, \ell$. Further, for $(z, p) \notin \mathrm{dom}\, \ell$, we have $\partial_1 \ell(z, p) = \emptyset$, which is closed, too. Therefore, $(z, p) \mapsto \partial_1 \ell(z, p)$ is closed-valued. Finally, since $(z, p) \mapsto \partial_1 \ell(z, p)$ is also outer semi-continuous, Rockafellar & Wets (1998, Exercise 14.9, p.649) implies that $\partial_1 \ell$ is measurable w.r.t. $\mathcal{B}(\mathcal{X} \times \mathcal{P})$. $\qquad\square$

**Corollary E.3.** *Suppose Assumption 6.7 holds. Then there exists a measurable selection for $\partial_1 \ell$, that is, a measurable map $v : \mathrm{dom}\, \partial_1 \ell \to \mathcal{X}$, such that $v(z, p) \in \partial_1 \ell(z, p)$ for all $(z, p) \in \mathcal{X} \times \mathcal{P}$.*

*Proof.* By Lemma E.2, the map $(z, p) \mapsto \partial_1 \ell(z, p)$ is closed-valued and measurable. Hence, the result follows directly from Rockafellar & Wets (1998, Corollary 14.6, p.647). $\qquad\square$

Now, we can prove Lemma 7.3:

*Proof.* Again, we can restrict to $b \in \mathbb{Q} \cap (0, \infty) =: \mathbb{Q}_+$. Thus, $\mathsf{A}_{\mathrm{err}}$ can be written as:

$$\mathsf{A}_{\mathrm{err}} = \left( \bigcup_{b \in \mathbb{Q}_+} \bigcap_{t \in \mathbb{N}_0} \mathsf{B}_{b,t} \right) \cap \left( \bigcap_{t \in \mathbb{N}_0} \{ (z^{(t)}, p) \in \mathrm{dom}\, \partial \ell \} \right) ,$$

where $\mathsf{B}_{b,t}$ is given by:

$$\mathsf{B}_{b,t} := \left\{ (p, (z^{(t)})_{t \in \mathbb{N}_0}) \in \mathcal{P} \times \mathcal{X}^{\mathbb{N}_0} \; : \; \| v(z^{(t+1)}, p) \| \le b \| z^{(t+1)} - z^{(t)} \| \right\} .$$

Hence, since $\sigma$-algebras are stable under countable unions/intersections, we only have to show measurability of the sets $\mathsf{B}_{b,t}$ and $\{ (z^{(t)}, p) \in \mathrm{dom}\, \partial_1 \ell \}$. Here, it holds that:

$$\{ (z^{(t)}, p) \in \mathrm{dom}\, \partial_1 \ell \} = (\Phi \circ (id, \pi_t))^{-1} (\mathrm{dom}\, \partial_1 \ell) ,$$

where $id$ is the identity on $\mathcal{P}$, and $\Phi : \mathcal{P} \times \mathcal{X} \to \mathcal{X} \times \mathcal{P}$ just interchanges the coordinates. By Lemma E.2, $\mathrm{dom}\, \partial_1 \ell$ is measurable, such that $\{ (z^{(t)}, p) \in \mathrm{dom}\, \partial_1 \ell \}$ is measurable for each $t \in \mathbb{N}_0$. Thus, it remains to show the measurability of the set $\mathsf{B}_{b,t}$. For this, introduce the function $g_b : (\mathrm{dom}\, \partial_1 \ell)^2 \to \mathbb{R}$, $((z_1, p_1), (z_2, p_2)) \mapsto \| v(z_2, p_2) \| - b \| z_2 - z_1 \|$. Since $v$ is measurable, and the norm is continuous, we have that $g_b$ is measurable. With this, we can write the set $\mathsf{B}_{b,k}$ as:

$$\begin{aligned} \mathsf{B}_{b,t} &= \{ g_b(z^{(t)}, p, z^{(t+1)}, p) \le 0 \} \\ &= \{ (g_b \circ (\pi_t, id, \pi_{t+1}, id) \circ \iota)(p, (z^{(t)})_{t \in \mathbb{N}_0}) \le 0 \} \\ &= (g_b \circ (\pi_t, id, \pi_{t+1}, id) \circ \iota)^{-1} (-\infty, 0] , \end{aligned}$$

where $\iota : \mathcal{P} \times \mathcal{X}^{\mathbb{N}_0} \to (\mathcal{X}^{\mathbb{N}_0} \times \mathcal{P})^2$ is the diagonal inclusion $(z_1, z_2) \mapsto ((z_2, z_1), (z_2, z_1))$, which again is measurable. Thus, $\mathsf{B}_{b,t}$ is measurable for each $t \in \mathbb{N}_0$ and $b \in \mathbb{Q}_+$, which concludes the proof. $\qquad\square$

## F. Proof of Lemma 7.4

*Proof.* By definition of the product $\sigma$-algebra on $\mathcal{P} \times \mathcal{X}^{\mathbb{N}_0}$, it suffices to show that $\tilde{\mathsf{A}}_{\mathrm{bound}}$ is measurable. Then, as it suffices to consider $c \in [0, \infty) \cap \mathbb{Q} =: \mathbb{Q}_+$, one can write $\tilde{\mathsf{A}}_{\mathrm{bound}}$ as:

$$\tilde{\mathsf{A}}_{\mathrm{bound}} = \bigcup_{c \in \mathbb{Q}_+} \bigcap_{t \in \mathbb{N}_0} \underbrace{\{ (z^{(t)})_{t \in \mathbb{N}_0} \in \mathcal{X}^{\mathbb{N}_0} \; : \; \| z^{(t)} \| \le c \}}_{=: \mathsf{C}_{c,t}} .$$

Thus, by the properties of a $\sigma$-algebra, it suffices to show that the sets $\mathsf{C}_{c,t}$ with $c \in \mathbb{Q}_+$ and $t \in \mathbb{N}_0$ are measurable. By defining $g(z) = \| z \|$, this follows directly from the identity $\mathsf{C}_{c,t} = (g \circ \pi_t)^{-1} [0, c]$. $\qquad\square$

## G. Architecture of the Algorithm for Quadratic Problems

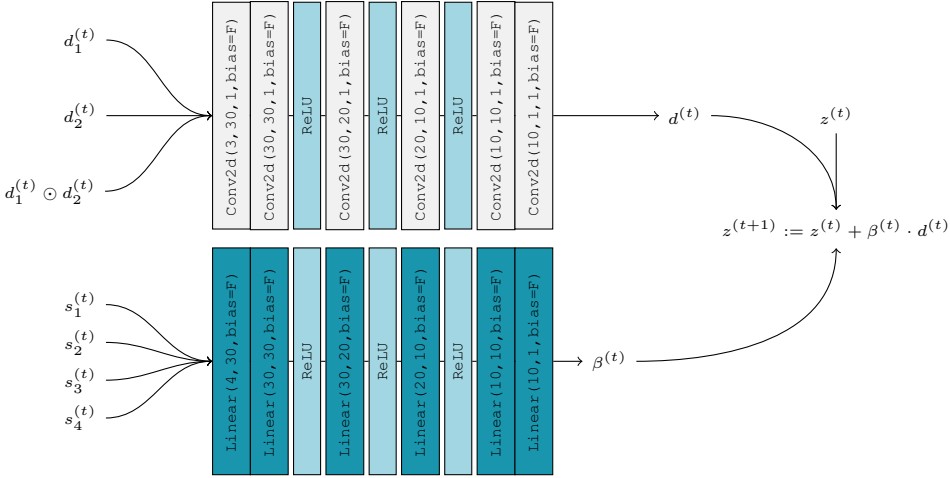

*Figure 3.* Update step of $\mathcal{A}$: The directions $d_1^{(t)}$, $d_2^{(t)}$ and $d_1^{(t)} \odot d_2^{(t)}$ are inserted as different channels into the `Conv2d`-block, which performs $1 \times 1$ "convolutions", that is, the algorithm acts coordinate-wise on the input. The scales $s_1^{(t)}, ..., s_4^{(t)}$ get transformed separately by the fully-connected block.

The algorithmic update is adopted from Sucker & Ochs (2024) and consists of two blocks:

1) The first block consists of $1 \times 1$-convolutional layers with ReLU-activation functions and computes the update direction $d^{(t)}$. As features, we use the normalized gradient $d_1^{(t)} := \frac{\nabla \ell(z^{(t)}, p)}{\|\nabla \ell(z^{(t)}, p)\|}$, the normalized momentum term $d_2^{(t)} := \frac{z^{(t)} - z^{(t-1)}}{\|z^{(t)} - z^{(t-1)}\|}$, and their coordinate-wise product $d_1^{(t)} \odot d_2^{(t)}$. The normalization is done to stabilize the training.

2) The second block consists of linear layers with ReLU-activation functions and computes the step-size $\beta^{(t)}$. As features, we use the (logarithmically transformed) gradient norm $s_1^{(t)} := \log\left(1 + \|\nabla \ell(z^{(t)}, p)\|\right)$, the (logarithmically transformed) norm of the momentum term $s_2^{(t)} := \log\left(1 + \|z^{(t)} - z^{(t-1)}\|\right)$, and the current and previous (logarithmically transformed) losses $s_3^{(t)} := \log\left(1 + \ell(z^{(t)}, p)\right)$, $s_4^{(t)} := \log\left(1 + \ell(z^{(t-1)}, p)\right)$. Again, the logarithmic scaling is done to stabilize training. Here, the term "+1" is added to map zero onto zero.

Importantly, we want to stress that the algorithmic update is not constrained in any way: the algorithm just predicts a direction and a step-size, and we do not enforce them to have any specific properties.

## H. Training of the Algorithm

For training, we mainly use the procedure proposed (and described in detail) by Sucker et al. (2024); Sucker & Ochs (2024). For completeness, we briefly summarize it here: In the outer loop, we sample a loss-function $\ell(\cdot, p)$ randomly from the training set. Then, in the inner loop, we train the algorithm on this loss-function with $\ell_{\text{train}}$ given by

$$\ell_{\text{train}}(h, p, z^{(t)}) = \mathbb{1}\{\ell(z^{(t)}, p) > 0\} \frac{\ell(z^{(t+1)}, p)}{\ell(z^{(t)}, p)} \cdot \mathbb{1}_{\mathsf{C}^c}(p, z^{(t)}) \,,$$

where $\mathsf{C} := \{(p, z) \in \mathscr{P} \times \mathcal{Z} : \ell(z, p) < 10^{-16}\}$ is the convergence set. That is, in each iteration the algorithm computes a new point and observes the loss $\ell_{\text{train}}$, which is used to update its hyperparameters. We run this procedure for $150 \cdot 10^3$ iterations. This yields hyperparameters $h^{(0)} \in \mathcal{H}$, such that $\mathcal{A}(h^{(0)}, \cdot, \cdot)$ has a good performance. However, typically, it is not a descent method yet, that is, $\mathbb{P}_{(P, \xi)|H = h^{(0)}}\{\mathsf{A}\}$ is small, such that the PAC-bound would be useless. Therefore, we employ the probabilistic constraining procedure proposed (and described in detail) by (Sucker et al., 2024) in a progressive way: Starting from $h^{(0)}$, we try to find a sequence of hyperparameters $h^{(1)}, h^{(2)}, ...$, such that

$$\mathbb{P}_{(P, \xi)|H = h^{(0)}}\{\mathsf{A}\} < \mathbb{P}_{(P, \xi)|H = h^{(1)}}\{\mathsf{A}\} < \mathbb{P}_{(P, \xi)|H = h^{(2)}}\{\mathsf{A}\} < ... \,.$$

*Remark* H.1. The notation $\mathbb{P}_{(P,\xi)|H}\{\mathsf{A}\}$ is not entirely correct and is rather to be understood suggestively, as the final prior distribution $\mathbb{P}_H$ is yet to be constructed. However, we think that it is easier to understand this way and therefore allow for this inaccuracy.

For this, we test the probabilistic constraint every 1000 iterations, that is: Given $h^{(i)}$, we train the algorithm (as before) for another 1000 iterations, which yields a candidate $\tilde{h}^{(i+1)}$. If $\mathbb{P}_{(P,\xi)|H=h^{(i)}}\{\mathsf{A}\} < \mathbb{P}_{(P,\xi)|H=\tilde{h}^{(i+1)}}\{\mathsf{A}\}$, we accept $h^{(i+1)} := \tilde{h}^{(i+1)}$, otherwise we reject it and start again from $h^{(i)}$. This finally yields some hyperparameters $h_0$ that have a good performance and such that $\mathbb{P}_{(P,\xi)|H=h_0}\{\mathsf{A}\}$ is large enough (here: about 90%). Then, starting from $h_0$, we construct the *actual* discrete prior distribution $\mathbb{P}_H$ over points $h_1, ..., h_{n_{\text{sample}}} \in \mathcal{H}$, by a sampling procedure. Finally, we perform the (closed-form) PAC-Bayesian optimization step, which yields the posterior $\rho^* \in \mathcal{M}_1(\mathbb{P}_H)$. In the end, for simplicity, we set the hyperparameters to

$$h^* = \underset{i=1,...,n_{\text{sample}}}{\arg\max} \ \rho^*\{h_i\}\,.$$

For the construction of the prior, we use $N_{\text{prior}} = 500$ functions, for the probabilistic constraint we use $N_{\text{val}} = 500$ functions, and for the PAC-Bayesian optimization step we use $N_{\text{train}} = 250$ functions, all of which are sampled i.i.d., that is, the data sets are independent of each other.

*Remark* H.2. Training the algorithm to yield a good performance is comparably easy. On the other hand, turning it into an algorithm, such that $\mathbb{P}_{(P,\xi)|H}\{\mathsf{A}\}$ is large enough (in our case: a descent method without enforcing it geometrically) is challenging and, unfortunately, not guaranteed to work. Nevertheless, it is key to get a meaningful guarantee.

## I. Architecture of the Algorithm for Training the Neural Network

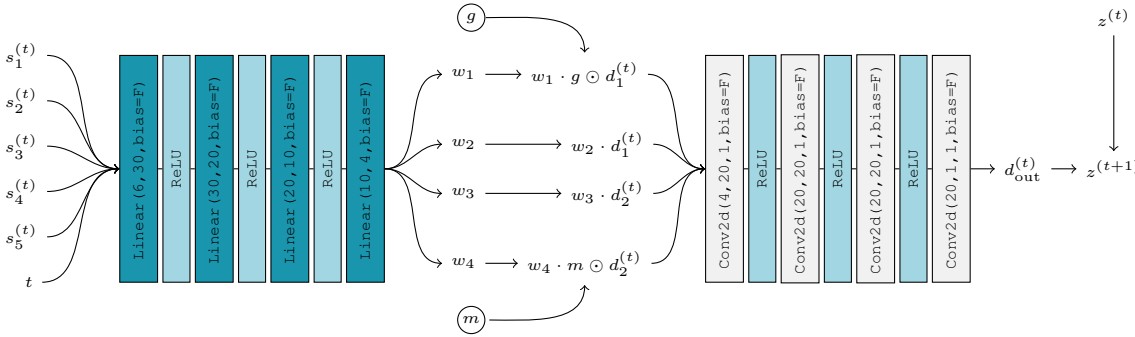

*Figure 4.* Algorithmic update for training the neural network: Based on the given six features, the first block computes four weights $w_1, ..., w_4$, which are used to perform a weighting of the different directions $g \odot d_1^{(t)}, d_1^{(t)}, d_2^{(t)}, m \odot d_2^{(t)}$, which are used in the second block. This second block consists of a 1x1-convolutional blocks, which compute an update direction $d_{\text{out}}^{(t)}$. Then, we update $z^{(t+1)} := z^{(t)} + d_{\text{out}}^{(t)}/\sqrt{t}$.

The algorithmic update is adopted from Sucker & Ochs (2024) and consists of two blocks:

1) The first block consists of linear layers with ReLU-activation functions and computes four weights $w_1, ..., w_4$. As features, we use the (logarithmically transformed) gradient norm $s_1^{(t)} := \log\left(1 + \|\nabla\ell(z^{(t)}, p)\|\right)$, the (logarithmically transformed) norm of the momentum term $s_2^{(t)} := \log\left(1 + \|z^{(t)} - z^{(t-1)}\|\right)$, the difference between the current and previous loss $s_3^{(t)} := \ell(z^{(t)}, p) - \ell(z^{(t-1)}, p)$, the scalar product between the (normalized) gradient and the (normalized) momentum term $s_4^{(t)}$, the maximal absolute value of the coordinates of the gradient $s_5^{(t)}$, and the iteration counter $t$.

2) The second block consists of $1 \times 1$-convolutional layers with ReLU-activation functions and computes the update direction $d_{\text{out}}^{(t)}$. As features, we use the normalized gradient $d_1^{(t)} := \frac{\nabla\ell(z^{(t)}, p)}{\|\nabla\ell(z^{(t)}, p)\|}$, the normalized momentum term $d_2^{(t)} := \frac{z^{(t)} - z^{(t-1)}}{\|z^{(t)} - z^{(t-1)}\|}$, and their "preconditioned" versions $g \odot d_1^{(t)}$ and $m \odot d_2^{(t)}$, where the weights $m, d \in \mathbb{R}^d$ are learned, too.

Again, we want to stress that the algorithmic update is not constrained in any way: the algorithm just predicts a direction, and we do not enforce them to have any specific properties.

