# OpenReview forum: "A Generalization Result for Convergence in Learning-to-Optimize"
_ICML.cc/2025/Conference — ICML 2025 oral_

### Official Review · Reviewer_685S · 2025-03-10

**Overall Recommendation:** 4

**Summary:**

This paper proposes a new method for analyzing the generalization ability of learning to optimize (L2O). The authors aim to formulate the convergence of L2O to unseen data as a random event measured by a posterior distribution of neural network's (NN) parameters. By assuming training ensures that the L2O generates a perfect convergence sequence with fast convergence and bounds on solutions, the authors bound the random event's probability by the probability of the convergence event on training data and the KL-divergence between the posterior and prior distributions of NN's parameters before and after training. Experimental results on synthetic test data demonstrate its generalization ability.

## update after rebuttal
I increase my final recommendation (from 2 to 4) due to the following reason:
1. This paper proposes a novel probabilistic perspective to demonstrate the generalization ability of L2O.
2. Although the proposed method imposes some strict conditions on training, the scheme shows that generalization ability can be guaranteed through well-designed training. The success of LLM seems to prove this to some extent.

**Claims And Evidence:**

No. This paper is quite hard to follow. Most technical details are not clearly introduced.

**Essential References Not Discussed:**

No

**Experimental Designs Or Analyses:**

Yes. The performance of the proposed method is a little better than Adam's, which may be much worse than other state-of-the-art, e.g., Math-L2O ICML 2023.

**Methods And Evaluation Criteria:**

No. The optimization problem (i.e., quadratic programming) in the experiment is too simple, which is insufficient to demonstrate that critical condition that training can be easily achieved. For example, non-convex and large-scale optimization problems, e.g., interference reduction problems in wireless communication, are more challenging and harder to converge even in training.

**Other Comments Or Suggestions:**

NA

**Other Strengths And Weaknesses:**

Strength:
1. This paper proposes a new paradigm for generalization demonstration of learning to optimize (L2O). The generalization is formulated as the random event of L2O's convergence on unseen data. The ability of generalization is measured by the posterior probability of convergence. Moreover, the probability is upper bound by the convergence of training and the KL divergence between posterior and prior distributions after and before training.

Weakness:
1. The effectiveness of the proposed method is limited. First, the main theorem, i.e., theorem 7.6 is loose, where the KL-divergence term illustrates the differences in the parameter distribution after and before training, which will be large unless training leads to little change from random initialization. Second, the conditions in lemmas 7.2-7.4 that the theorem relies on are too strict. For example, lemmas 7.2 and 7.3 require the trained L2O to converge fast and lemma 7.4 requires the solution generated by L2O to be bounded by a constant. These conditions are more than the empirical convergence of training but also additionally require a specific shape of sequences generated by L2O, which may be hard to satisfy in practice.

**Questions For Authors:**

1. Can you give some explanation for the claim that most functions satisfy KL inequality?
2. Can you demonstrate how training ensures lemmas 7.2-7.4?

**Relation To Broader Scientific Literature:**

This paper aims to give a general formulation of the generalization of learning-to-optimize with a probabilistic method. The method is related to the PAC-Bayesian learning theory and the paper Sucker & Ochs (2024). The method is also related to the generalization of neural networks, which is a hot topic in the field of machine learning.

**Theoretical Claims:**

No.

---

> ### Author Rebuttal · Authors · 2025-03-31
>
> We would like to thank the reviewer for taking the time to provide this feedback, even it was unfortunately rather negative.
>
> Regarding the claim that the paper is hard to follow and that most technical details are not clearly introduced:
> 1) Could you please be more specific here? Otherwise, we cannot improve the paper.
> 2) In principle, we agree with you that our paper is not straightforward. However, we think that this is rather due to the fact that it combines many advanced and abstract concepts (conditional distributions over the space of trajectories). This is why we have explained our main idea in a separate section, such that it does not get obscured by the technical details. On the other hand, we kindly disagree with the fact that we did not introduce the technical details: We devoted half a page for the notation, and another 1.5 pages just to explain the setup. Besides that, we have provided basically all other needed notions in the appendix or defined them in the text.
>
> Regarding methods, evaluation criteria and the experiments:
> 1) We do not claim that training the algorithm in such a way that it satisfies the properties of our theorem is actually easy. In fact, we explained in the conclusion that there are cases where it is hard. Since our setting is very abstract and applies to many optimization algorithms, there will probably always be cases where training is hard, and future research should address how to make this simpler.
> 2) We also train a neural network, that is, a non-smooth non-convex optimization problem.
> 3) While it is true that the results are only a bit better than Adam, this does not apply to the other experiment. Additionally, the main contribution of this paper are not the experiments; they are just to showcase the validity of our theoretical claims, that is, the generalization and thus convergence of the methods.
>
> Regarding your posed weakness: We agree that there are limitations to the PAC-Bayesian approach. However, these are well-known and people have come up with ways how to circumvent them, for example data-dependent priors. We did not comment on it, because it is well-known and not the actual scope of our paper. \
> We partly agree and partly disagree with the claim that the used conditions are too strict:
> 1) Since these conditions are based on the theorem due to Attouch et al., you basically just claim that this theorem uses unreasonably restrictive assumptions. Additionally, please note that our proof-strategy can be combined with different theorems with potentially milder assumptions!
> 2) It is true that Lemmas 7.2 and 7.3 are related to the convergence rate of the algorithm. However, most of the time a faster convergence of the learned algorithm is the only reason why we use learning-to-optimize. Thus, claiming that fast convergence is problematic seems not appropriate.
> 3) We agree that the boundedness condition is problematic. However, this is a common problem also for many conventional optimization algorithms, that is, even the convergence proofs of many conventional optimization algorithms rely on boundedness, such that we do not expect to solve it in this more difficult setting here.
> 4) We might be mistaken, but it feels like there is a certain confusion: You claim that “These conditions are more than the empirical convergence of training but also additionally require a specific shape of sequences generated by L2O, which may be hard to satisfy in practice.”. Please note that there is a subtle yet decisive problem: *Often you simply cannot observe convergence empirically.* This is due to the fact that convergence is an asymptotic notion and therefore, by definition, not observable in practice. Thus, if we want to judge about it from observations, we need to change the perspective and try to observe properties (the "specific shape" of the trajectory) that allow for *deducing* convergence. And this is accomplished by our theorem.
>
> We would like to point out again that, while these properties are used in our main theorem, a large part of our contribution is actually the *proof-strategy*, which allows to derive similar theorems under different assumptions.
>
> Regarding your questions:
> 1) E.g, semi-algebraic functions satisfy the KL-inequality. These are “functions whose epigraph can be written as a finite union of sets, each defined by finitely many polynomial inequalities”, which is a very large class of functions. These are the “prototypical nonpathological functions
> in nonsmooth optimization” (Drusvyatskiy et al., ”Curves of descent”). Even more generally, functions that are definable in an o-minimal structure or that are tame do satisfy the KL-inequality (for example, see the paper by Attouch et al.).
> 2) We do not claim that training necessarily ensures these conditions. Actually,sometimes it is hard, which is why future research should solve this. Anyways, one could try an explicit optimization of the parameters a and b.

---

> > ### Comment · Reviewer_685S · 2025-04-03
> >
> > Thank you for your detailed feedback.
> >
> > My main concern is for the triviality of the conditions proposed in Lemmas 7.1-7.4 (also Theorem 6.5). Although the presented experiment on meta-train a L2O model to train a $L_2$-norm regression problem with NN, the target NN is too small that its convergence is easy to achieve. However, in general, L2O does not always win. Otherwise, conventional solvers will no longer be used. For example, the famous LISTA [Lecun et al., ICML 2010] still suffers poor convergence on high-dimensional problems.
> >
> > Moreover, two counterexamples in Appendix B are not sufficient to demonstrate it is non-trivial.  I am not sure whether it is a typo or not, the LHS is on iteration $^{(t+1)}$ but not $^{(t)}$. One can easily verify that gradient descent also may violate the condition for some sequences.
> >
> > However, I agree with the author's statement about the contribution. The rating is updated.

---

### Official Review · Reviewer_zRws · 2025-03-12

**Overall Recommendation:** 4

**Summary:**

Learn-to-optimize has been a popular research topic in recent years.
However, many theoretical guarantees are still lacking.
This paper develops a probabilistic framework that resembles
classical optimization and allows for transferring geometric
arguments into learn-to-optimize. The paper establishes
a generalization result for very general loss functions and shows
convergence of learned optimization algorithm to critical points
with high probability. Numerical experiments
are provided for solving a quadratic problem and training neural networks.

**Claims And Evidence:**

Yes.

**Essential References Not Discussed:**

N.A.

**Experimental Designs Or Analyses:**

Yes.

**Methods And Evaluation Criteria:**

Yes.

**Other Comments Or Suggestions:**

(1) In Theorem 6.3. and later Theorem 7.6., $\Phi_{a}^{-1}(p):=\frac{1-\exp(-ap)}{1-\exp(-a)}$ plays a central role.
It would be nice if you can provide some intuition explanations what this function $\Phi_{a}^{-1}(p)$
is about and where it comes from.

(2) In the references, please be more consistent. For example, In Langford and Caruana, the booktitle is capitalized,
but not in Langford and Shawe-Taylor. In addition, the book name by Nesterov should be capitalized.

(3) In the proof of Lemma 7.1., you wrote $B_{\varepsilon}(p,z)$ ult. What is ult.?

**Other Strengths And Weaknesses:**

Strengths

(1) The paper is well-written.

(2) The theory is accompanied by experiments.

(3) The theory is rigorously derived by paying a lot of attention to the technical details.
For example, the paper devotes an entire Section 7.1. to show measurability condition
holds in order to apply some existing results. A measurability condition is often taken
as granted. But the paper digs deep into checking this technical condition, and provides
a very details and non-trivial proof in the Appendix.

Weaknesses

(1) The proofs of the main results seem to be direct consequence by applying
Theorem 6.3. and Theorem 6.5 that are both available from the recent literature.
This makes the paper seem to be an application built upon the very recent literature
and makes the contributions less significant.

(2) The paper mentions four weaknesses in the conclusion section, which is fair and honest.
I think some of the weaknesses seem to be inherent. But some might be overcome which
can be left as future research directions.

**Questions For Authors:**

N.A.

**Relation To Broader Scientific Literature:**

Learn-to-optimize has been a popular research topic in recent years.
However, many theoretical guarantees are still lacking.
This paper develops a probabilistic framework that resembles
classical optimization and allows for transferring geometric
arguments into learn-to-optimize to fill in this gap.

**Theoretical Claims:**

The paper seems to be correct though I didn't check the proofs in details.

---

> ### Author Rebuttal · Authors · 2025-03-31
>
> We would like to thank the reviewer for giving this feedback, and we would like to take the opportunity to shortly comment on the weaknesses and to answer the questions.
>
> Regarding your first posed weakness: It is true that, on an abstract level, in the end the result follows by combining Theorems 6.3 and 6.5. However, we think that it is still a significant step, because:
> 1) There is a very subtle problem with convergence results: In general, convergence is an asymptotic notion, that is, it belongs to the so-called tail-$\sigma$-algebra and therefore is *simply not observable* in practice. Thus, if we want to judge about convergence based on observations during training, we need to observe properties that allow to *deduce* convergence again. And this is accomplished by our new proof-strategy.
>  2) You have to know both results, which are both rather abstract and advanced in their respective fields (PAC-Bayes and non-smooth non-convex optimization), and you have to combine them in the correct way, that is, you have to see that all resulting properties have to be phrased as sets in the space of trajectories. Since we have done it now here, it will also be comparably easy to adapt for follow-up work.
> 4) You have to do the proofs, which are by no means trivial in this abstract setting dealing with
> spaces of sequences.
> Lastly, we want to point out that this proof-strategy is actually also part of our contribution. It bridges the gap between conventional optimization theory and generalization bounds. And finding a completely new proof-strategy is often harder than adapting an existing one to a new problem.
>
> Regarding your second posed weakness: We do not fully understand why you regard this as a weakness of our paper? Could you elaborate on that?
>
> Regarding your comments:
> 1) The function $\Phi_a$ is related to the log-Laplace transform of a Bernoulli random variable (see Catoni (2007)). This in turn is related to the moment-generating function, which can be used to characterize how the random variables concentrates. And this is commonly used to get generalization
> results. $\Phi_a$ and $\Phi_a^{-1}$ a were both introduced like this by Catoni (2007). \
> 2) Thanks for the hint! We will check and update the references. \
> 3) “ult.” is an abbreviation for “ultimately” and refers to the fact that the sequences only ultimately have to lie in the ball with radius ϵ, that is, asymptotically (this is why we have this union/intersection over all iterates in the definition). This notation is used in analogy to the notation from probability theory (for example, see Kallenberg).

---

### Official Review · Reviewer_FnHp · 2025-03-13

**Overall Recommendation:** 4

**Summary:**

While learning-to-optimize has shown to be a powerful paradigm to enhance the efficiency of the optimisation phase for problems similar to the one encountered during training, it is unsure how such a trained algorithm will behave on unseen problems with different internal structure. This work tackles this issue by proposing a novel PAC-Bayes generalization bound for the learning-to-optimize problem when the considered learning algorithm possesses a Markovian structure and that the (potentially non-smooth, non-convex) loss satisfies the Kurdyka-Lojasiewicz condition.

**Claims And Evidence:**

This is a truly solid mathematical paper, with a well-explained and convincing proof-strategy. I read the proof of measurability of Section 7 which are well-written, sound and rigorous.

**Essential References Not Discussed:**

I do not know enough Sucker & Ochs 2024 and the associated references to be aware of the SOTA in the links between PAC-Bayes and learning-to-optimize.

**Experimental Designs Or Analyses:**

No.

**Methods And Evaluation Criteria:**

I did not check the experimental protocol, however the nature of the problem (quadratic or small neural networks) look reasonable.

**Other Comments Or Suggestions:**

None.

**Other Strengths And Weaknesses:**

None, this is a good theoretical paper involving an innovative use of PAC-Bayes. However, I may overestimate the novelty of those techniques as I was not aware of Sucker&Ochs 2024

**Questions For Authors:**

- Would it be possible to discuss more the influence of the prior in Theorem 6.3 and 7.6?
- The reference (Theorem 42, Sucker&Ochs 2024) for theorem 6.3 does not seem correct, can you update it?
- In Theorem 7.6 is it possible to implement the sum $\frac{1}{N}\sum_{i=1}^N \mathbb{P}\_{(P,\xi)| H}(A^c)$ can you expand on it?

**Relation To Broader Scientific Literature:**

Could the authors expand a bit more on the comparison between your results and those of Sucker&Ochs 2024, which looks quite close according to what you say. Would it be possible to state such a theorem in the appendix to highlight the originality of your contribution?

**Theoretical Claims:**

Once the setup of measurable events is well-posed and proof of the main-results are a quite straightforward combination of results of Catoni 2007 and Attouch et al. 2013.

---

> ### Author Rebuttal · Authors · 2025-03-31
>
> Also here, we would like to thank the reviewer for taking the time to provide this feedback. We are glad that you considered the proof-strategy as “well-explained”, because it is one of our main contributions.
>
> Regarding your question whether we could expand a bit more on the comparison between our results and those of Sucker&Ochs (2024): First, while Sucker&Ochs state Theorem 42, they only use it for a small remark, where they explain that this result can be used to bound a function r that is used to estimate something like a linear rate of convergence. If one would perform a similar analysis
> as we have done here for the case that a) the loss function has a unique minimizer and b) the algorithm does indeed converge with a linear rate, this could be used to deduce convergence, which, however, Sucker&Ochs also left completely unaddressed. \
> Furthermore, this setting would be a very specific case in which you can get convergence by just observing a certain function-value. In general, however, this is not the case, due to the following subtle problem: Convergence is an asymptotic notion, such that it belongs to the so-called tail-$\sigma$-algebra, that is, in practice it is *inherently non-observable*. Thus, if we want to judge about convergence from observations during training, we need to switch the perspective and try to observe properties, that allow to *deduce* convergence again. And this is what our proof-strategy accomplishes. Although this difference is very subtle, it is still crucial. For L2O this applies in particular to the non-smooth and non-convex case, because the (sub-)gradient does not necessarily say anything about the distance to a critical point. \
> Additionally, we would like to point out that part of our contribution is also to show how we can actually combine results from conventional optimization theory with generalization results, such that it can be adapted to other cases quite easily.
>
> Regarding the follow-up question of whether we could state such a theorem in the appendix to highlight the originality of our contribution: What exactly do you mean by “such a theorem”? Theorem 6.3 is the theorem by Sucker&Ochs.
>
> Regarding the discussion of the prior: The prior is highly important, because the generalization bound is more tight, if the posterior stays close to prior (KL-term is smaller). On the other hand, this means that the prior should already yield some reasonable
> performance. Therefore, a well-known remedy is that one typically has a two-step learning process: First, the prior gets optimized to yield a reasonably good performance and then the posterior gets chosen, which provides the guarantee.
> We did not comment on it, because it is not the actual contribution of our paper and the approach is already well-known.
>
> Regarding your comment that the reference (Theorem 42, Sucker& Ochs 2024) for Theorem 6.3 seems to be wrong: We think it is correct. Maybe it is confusing, because Sucker&Ochs are again referencing to Catoni (2007). However, we think that this is just to acknowledge that their result turns out to be basically the same as the one by Catoni (who considered Bernoulli random variables).
>
> Regarding your question for Theorem 7.6: In our case, the conditional probability is simply an indicator-function (note that in your comment there is a $P_n$ missing!), because the algorithm is deterministic and everything else is given from the data. So it turns into an empirical mean over indicator functions. However, we did state it in this more general version, because the proof-idea could also be used for stochastic algorithms, in which case one would have to estimate the probability, for example over several runs of the algorithm.

---

### Official Review · Reviewer_vfny · 2025-03-14

**Overall Recommendation:** 4

**Summary:**

This paper presents a probabilistic framework to establish convergence guarantees for L2O algorithms, addressing the challenge that conventional geometric arguments for convergence do not readily apply to learned optimizers. The key contribution is a generalization result that combines PAC-Bayesian learning theory with variational analysis, specifically leveraging the KL inequality, to show that L2O algorithms converge to critical points with high probability. The authors develop a novel proof strategy that translates worst-case convergence analysis into a probabilistic setting, removing the need for restrictive safeguard mechanisms in algorithm design. Experimental results on quadratic optimization problems and neural network training demonstrate that the learned optimizers outperform traditional methods such as heavy-ball acceleration and Adam, while the PAC-Bayesian framework provides nontrivial guarantees on their generalization.

## update after rebuttal
The author's response has persuaded me that this is strong work, so I'm increasing my score from 3 to 4.

**Claims And Evidence:**

Yes.

**Essential References Not Discussed:**

No.

**Experimental Designs Or Analyses:**

The experiments appear to be sound.

**Methods And Evaluation Criteria:**

Yes.

**Other Comments Or Suggestions:**

N.A.

**Other Strengths And Weaknesses:**

Strengths

1. The paper tackles a really interesting and important problem—the convergence of L2O. This is an open question that very few papers have addressed, despite L2O’s growing popularity. Most work in this area has focused on empirical performance without theoretical guarantees, so I appreciate that this paper makes an effort to analyze convergence, which is a crucial missing piece in the field.

2. I also like the core idea of the paper—moving away from traditional geometric analysis, which often requires the learned optimizer to have specific properties that may not hold in practice. Since L2O relies on neural networks, their outputs are difficult to control, and enforcing geometric constraints can limit their flexibility. Instead, this paper uses a PAC-Bayesian framework combined with the KL inequality, which provides a more natural way to study convergence without imposing artificial restrictions. This makes the approach both theoretically interesting and practically relevant.

Weaknesses

1. One key assumption in this paper is that L2O converges on the training set, and the main focus is on whether this convergence generalizes to new optimization problems. While this assumption is reasonable and intuitive, it is itself an open question that remains unaddressed. The paper does not explore whether or under what conditions L2O is actually guaranteed to converge during training, which makes the overall analysis feel somewhat incomplete. I understand that this is beyond the scope of the paper, as the focus is on generalization rather than training dynamics, but leaving this assumption undiscussed does create some discomfort. This issue is particularly important because L2O training is not a trivial process. The convergence of NN training itself is already a complex and open problem, with theoretical frameworks like NTK providing some insights but requiring unrealistic infinite-width assumptions. L2O, being an even more complex NN that integrates both the optimization process and neural network prediction, presents a significantly harder convergence problem than standard architectures like MLPs, which are already difficult to analyze. Given this, the absence of a solid consensus on the convergence of L2O during training greatly weakens confidence in the overall framework presented in the paper.

2. The presentation of the paper could be significantly improved. The structure feels disorganized, making it harder to follow the key ideas and contributions. Some sections could be better structured to improve clarity, and certain explanations—especially in the theoretical parts—could be made more intuitive. A clearer organization would help readers grasp the core insights more effectively.

**Questions For Authors:**

N.A.

**Relation To Broader Scientific Literature:**

L2O has been widely studied as a means of leveraging machine learning to design optimization algorithms that adapt to specific problem structures, with earlier works focusing on empirical performance but lacking rigorous convergence guarantees. This paper advances the field by introducing a probabilistic framework that integrates PAC-Bayesian generalization theory with variational analysis to provide convergence guarantees in generalization. Prior works on PAC-Bayesian learning have established generalization bounds for learning-based methods, and KL-based convergence analysis has been used in traditional optimization, but their combination in the context of L2O is novel. Additionally, the paper addresses limitations of safeguard-based approaches, which constrain learned optimizers to fit classical convergence analyses, by formulating a more flexible probabilistic guarantee.

**Theoretical Claims:**

I did not verify all the proofs in detail, but based on a high-level review, they appear to be reasonable.

---

> ### Author Rebuttal · Authors · 2025-03-31
>
> We would like to thank the reviewer for taking the time to provide this detailed feedback.
> We shortly want to comment on the stated weaknesses:
>
> We agree that such convergence guarantees are highly desirable, but we also think that
> this is asking for too much: We are considering an abstract algorithm (think of a neural network) and an
> abstract loss function, and such an analysis is typically performed in a problemspecific
> way, even for conventional optimization algorithms. Thus, while we could
> provide hypothetical examples that explain our contributions for a concrete setting,
> we do not think that we can prove the convergence of such a learned
> method in general, also because it additionally is influenced by the used training
> procedure. Further, if we could perform such an analysis and provide the
> corresponding guarantees, we might be able to do the same for the test data,
> such that we would not even need the generalization anymore. However, the
> key idea is exactly that we acknowledge the fact that we are often not able to perform
> such an analysis, in which case we have to rely on observations during training. And for these cases we can apply our generalization result,
> which is widely applicable, exactly because our setting is so abstract. \
> Furthermore, how to train optimization algorithms
> so that they do satisfy these convergence-ensuring properties for most of the
> training data is actually also part of future research. \
> Lastly, we want to stress again that a large part of our contribution is also the *proof-strategy* which,
> after it has been established, can be applied or adapted quite easily to several
> different applications. Here, we want to emphasize one thing, because the problem is very subtle: For many applications, convergence of the learned algorithm is a so-called asymptotic event and belongs to the tail-$\sigma$ algebra, that is, by definition it is *practically impossible* to observe it directly. Thus, if we want to judge about it based on observations (like in L2O), we need to observe something that allows us to *deduce* it. And this is exactly what our strategy is exploiting.
>
> Regarding your claim that the presentation could be significantly improved:
> Could you please be a bit more concrete here? We are of course always interested
> in improving the clarity of our paper, but, like this, it is rather hard to see which
> parts should be restructured or presented differently, such that, for now, we would
> rather kindly disagree with this remark. For us, the paper has a clear inherent
> flow, which is the following:
> 1) Start of the paper (introduction/motivation + related work)
> 2) Presentation of main idea in a simplified way, so that readers can grasp it
> more easily.
> 3) Notation and background material, such that claims can be made rigorous.
> 4) Recalling main idea and concretizing it for the setting of learning-to-optimize
> (lemmas preparing the main result + main result)
> 5) Experiments

---

> > ### Comment · Reviewer_vfny · 2025-04-03
> >
> > Thank you for the detailed response. I think the paper would benefit from expanding Section 4 to better highlight the core idea in a more intuitive way, without too much mathematical detail. It would also be helpful to include more discussion on the convergence behavior of L2O during training, even if only at a high level. Overall, this is a solid contribution, and I have raised my score to 4.

---

### Decision · Program_Chairs · 2025-05-01

**Decision:**

Accept (oral)

**Comment:**

The paper brings a rare convergence guarantee analysis to the learning to optimized framework. This is achieved thanks to an innovative usage of the PAC-Bayesian theory (partly foreseen in recent work but yet unexploited) combined with non-convex optimization and stochastic processes literature.

The reviewers agreed that this is a strong mathematical paper, as summarized by Reviewer zRws:
> The theory is rigorously derived by paying a lot of attention to the technical details. For example, the paper devotes an entire [section] to show measurability condition holds in order to apply some existing results. A measurability condition is often taken as granted. But the paper digs deep into checking this technical condition, and provides a very details and non-trivial proof in the Appendix.

Despite being performed on simple problems, the provided experiments convincingly "showcase the validity of [the] theoretical claims".